# Triple-BERT: Do We Really Need MARL for Order Dispatch on Ride-Sharing Platforms?

## Abstract

On-demand ride-sharing platforms, such as Uber and Lyft, face the intricate real-time challenge of bundling and matching passengers—each with distinct origins and destinations—to available vehicles, all while navigating significant system uncertainties. Due to the extensive observation space arising from the large number of drivers and orders, order dispatching, though fundamentally a centralized task, is often addressed using Multi-Agent Reinforcement Learning (MARL). However, independent MARL methods fail to capture global information and exhibit poor cooperation among workers, while Centralized Training Decentralized Execution (CTDE) MARL methods suffer from the curse of dimensionality. To overcome these challenges, we propose Triple-BERT, a centralized method designed specifically for large-scale order dispatching on ride-sharing platforms. Built on TD3, our approach addresses the vast action space through an action decomposition strategy that breaks down the joint action probability into individual driver action probabilities. To handle the extensive observation space, we introduce a novel BERT-based network, where parameter reuse mitigates parameter growth as the number of drivers and orders increases, and the attention mechanism effectively captures the complex relationships among the large pool of driver and orders. We validate our method using a real-world ride-hailing dataset from Manhattan. Triple-BERT achieves approximately an 11.95% improvement over current state-of-the-art methods, with a 4.26% increase in served orders and a 22.25% reduction in pickup times. Our code, trained model parameters, and processed data are publicly available at the anonymous repository https://anonymous.4open.science/r/Triple-BERT.

## 1 Introduction

Ride-sharing platforms, such as Uber and Lyft, face the complex challenge of dynamically matching passengers with distinct origins and destinations to available vehicles in real time. This task must account for significant system uncertainties, including fluctuating demand, varying traffic conditions, and the availability of drivers. As the volume of concurrent ride requests increases, these platforms must efficiently allocate resources to minimize detours, reduce waiting times, and maximize customer satisfaction and platform revenue. However, the inherently large and dynamically changing action and observation spaces make this problem highly challenging for the operation of ride-sharing platforms.

Recently, Reinforcement Learning (RL) methods have shown great potential in addressing the order dispatching problem in ride-sharing platforms. Model-free RL, in particular, enables agents to autonomously learn optimal dispatching policies by interacting with the environment, without requiring complex system modeling. This approach allows platforms to optimize multiple objectives, including platform income, driver payments, and customer satisfaction. Despite these advantages,

applying RL to large-scale order dispatching introduces significant challenges. The vast action and observation spaces, stemming from the large number of drivers and orders, make sufficient exploration and efficient training difficult. Multi-Agent Reinforcement Learning (MARL) methods have been widely adopted to address these challenges by decomposing the problem into smaller subproblems for individual agents (drivers). Independent MARL methods, such as IDDQN [1; 2; 3] and ISAC [4], are computationally efficient but fail to capture global information and exhibit limited cooperation among agents. Graph Neural Networks (GNNs) have been introduced to enable the network to capture neighboring information for each agent, alleviating this issue to certain extent [5; 6]. Meanwhile, Centralized Training with Decentralized Execution (CTDE) methods, such as QMIX [7] and CoPO [8], struggle with the curse of dimensionality when applied to large-scale scenarios with thousands of agents, resulting in slow convergence and suboptimal performance.

To address these limitations, this paper proposes a centralized Single-Agent Reinforcement Learning (SARL) method, named Triple-BERT, tailored for large-scale order dispatching in ride-sharing platforms. Triple-BERT introduces an action decomposition method that simplifies the joint action probability into individual driver action probabilities, enabling each driver to make independent decisions while maintaining global coordination. The method leverages TD3 [9] for optimization, with modifications to the actor optimization process via policy gradient [10] to better suit the ride-sharing context. To handle the extensive observation space, we design a novel BERT-based [11] neural network architecture. This network employs bi-directional self-attention to effectively capture complex relationships between drivers and orders, while its parameter reuse mechanism prevents parameter explosion as the number of drivers and orders increases. Additionally, compared to MARL, SARL faces a unique challenge of sample scarcity, as the records of multiple agents are merged into a single training stream. To address this, we propose a two-stage training strategy, where feature extractors are pre-trained using a MARL approach to learn general embedding capabilities, followed by centralized fine-tuning. The main contributions of this paper can be summarized as follows:

- We introduce Triple-BERT, which is the first centralized SARL framework for large-scale order dispatching on ride-sharing platforms. This approach addresses the limitations of the observation space and the inefficiencies in cooperation among agents present in MARL methods. To tackle the large action space inherent in the matching problem of order dispatching tasks, we propose an action decomposition method that breaks down the joint action probability into individual driver action probabilities. Additionally, we propose a two-stage training method to address the sample scarcity issue in SARL, where the feature extractors are first trained using a MARL approach.

- To support the proposed RL framework in a large observation space, we develop a novel neural network architecture based on BERT. This design leverages self-attention mechanisms to effectively capture the relationships between drivers and orders. Furthermore, we incorporate a QK-attention module to reduce computational complexity from multiplication to addition in the order dispatching task, along with a positive normalization method to mitigate parameter redundancy issues.

- We validate the proposed method in the ride sharing scenario, using a real-world dataset of ride-hailing trip records from Manhattan. Our method outperforms the MARL methods reported in previous works, demonstrating approximately a 11.95% improvement over current state-of-the-art methods, with a 4.26% increase in served orders and reductions of about 22.25% in pickup time.

## 2 Problem Setup

In this paper, we address the order dispatching task within on-demand logistic systems, such as ride hailing, food delivery, and express delivery. We consider a platform managing $n$ drivers (hereafter referred to as workers), represented by the state $W_t = \{w_{1,t}, w_{2,t}, \ldots, w_{n,t}\}$, where $w_{i,t}$ denotes the state of worker $i$ at time $t$. At each time step, the platform processes a set of orders, including newly arrived orders and any previously unassigned orders, denoted as $O_t = \{o_{1,t}, o_{2,t}, \ldots, o_{m_t,t}\}$, where $m_t$ is the total number of orders at time $t$. Since real-time performance is crucial in on-demand systems, the platform aims to bundle and assign orders in a way that minimizes delivery time while maximizing the number of served orders. Customers are assumed to be impatient; if an order is not acknowledged within a specified time frame, workers will decline it. Moreover, late deliveries beyond the scheduled time may result in customer complaints, potentially causing losses for the platform. The overall workflow is illustrated in Fig. 1, and the Markov Decision Problem (MDP) is formulated as $< S, A, R, P >$, encompassing the state, action, reward, and transition function, which will be detailed below:

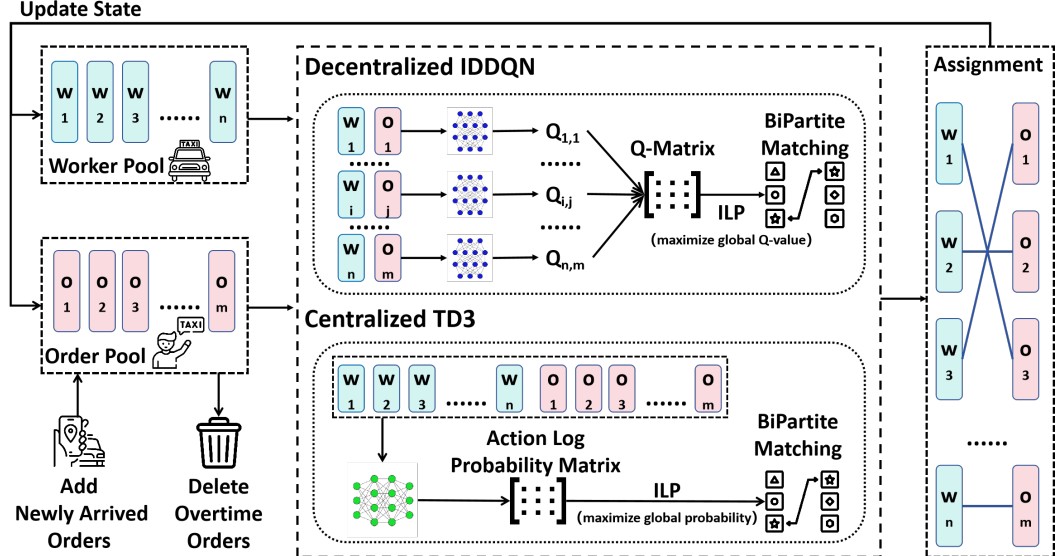

Figure 1: Workflow: At each time step, the worker and order pools update their states based on the assignments made in the previous time step. Specifically, the order pool adds newly arrived orders and removes overdue ones. For IDDQN, the Q-value of each worker-order pair is calculated, and ILP is applied to maximize the global Q-value. For TD3, the probability of each worker-order pair is computed, followed by the application of ILP to maximize the global assignment probability.

**(i) State:** At timestep $t$, the state or observation can be represented as $S_t = [W_t, O_t]$, consisting of the states of workers and orders. For the order $j$ to be assigned, the state $o_{i,j}$ includes the order's origin and destination, pickup time, and scheduled arrival time. For each worker $i$, the state $w_{i,t}$ consists of the onboard orders $H_{i,t}$ that are still unfinished, the current location, the residual capacity, and the estimated time when he/she will be available to accept a new order. (Note that we assume if a worker is en route to pick up a new order or if his/her capacity is full, he/she cannot serve a new order.) Specifically, $H_{i,t}$ is a sequence of orders $H_{i,t} = \{h_{i,1,t}, h_{i,2,t}, \ldots, h_{i,k_{i,t},t}\}$, where $k_{i,t}$ is the number of onboard orders for worker $i$ at time $t$ and each order $h_{i,k,t}$ contains the same information as the orders to be assigned $o_{j,t}$.

**(ii) Action:** At each time $t$, the action can be represented as $A_t = \{a_{1,t}, a_{2,t}, \ldots, a_{n,t}\}$, where each $a_{i,t}$ is an $m_t$-dimensional vector with at most one element set to 1, indicating which order is assigned to worker $i$. The order dispatching task is particularly challenging due to two main factors: (i) the size of the action space keeps changing over time because the number of orders $m_t$ varies dynamically as new orders arrive and old orders are completed or canceled; (ii) the size of the action space is extremely large for real systems. For instance, considering $n = 1000$ workers and $m_t = 10$ orders, the action space can reach approximately $10^{30}$. (A detailed proof is provided in Appendix A.) This combination of an enormous action space and its continuously changing size significantly complicates sufficient exploration and stable network convergence for standard RL methods.

**(iii) Reward Function:** We split the reward function for each worker, meaning each worker will receive a reward $r_{i,t+1}$ at time step $t$, and the global reward is the sum of each worker's reward: $R_{t+1} = \sum_{i=1}^{n} r_{i,t+1}$. The reward $r_{i,t+1}$ can be calculated according to the following function:

$$r_{i,t+1} = \mathcal{R}(s_{i,t}, a_{i,t}) = \begin{cases} \beta_1 + \beta_2 p_{i,t}^{in} - \beta_3 p_{i,t}^{out} - \beta_4 \chi_{i,t} - \beta_5 \rho_{i,t} \,, & |a_{i,t}| = 1 \\ 0 \,, & |a_{i,t}| = 0 \end{cases} \tag{1}$$

where $\beta_1$ to $\beta_5$ are non-negative weights representing the platform's valuation of each term, $p_{i,t}^{in}$ and $p_{i,t}^{out}$ represent the income from customers and the payout to workers, respectively. The variables $\chi_{i,t}$ and $\rho_{i,t}$ represent the number of en-route orders that will exceed their scheduled time and the additional travel time of all en-route orders when the assigned order is added to the scheduled route of worker $i$ at time $t$, respectively. This reward function is designed to comprehensively consider the interests of the platform, workers, and customers, mimicking the operation of a real-world food delivery platform. It is important to emphasize that $p_{i,t}^{in}$ and $p_{i,t}^{out}$ are calculated based on the order

120 distance and the additional travel distance for the worker, respectively. When calculating travel time,
121 we will utilize the Traveling Salesman Problem (TSP) to optimize the worker's route.

122 **(iv) Transition Function:** In our system, the reward is deterministic given the current state and action.
123 Therefore, the transition function is represented by $P(S_{t+1}|S_t, A_t)$. In this study, the transition
124 probabilities are not explicitly modeled; instead, they are inferred through the model-free RL.

## 3 Methodology

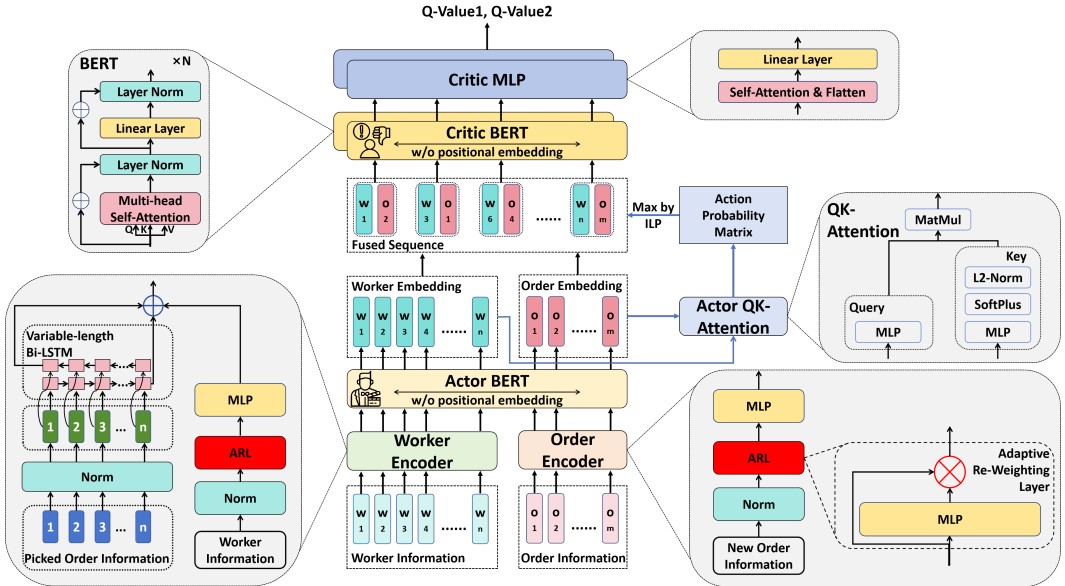

Figure 2: Proposed Network Architecture: In this figure, the fused sequence (input to Critic-BERT) represents workers $1$, $3$, $6$, and $n$ selecting orders $2$, $3$, $4$, and $m$, respectively.

### 3.1 Overview

127 In this work, we aim to utilize centralized SARL to address the large-scale order dispatching task,
128 with the goal of enabling the model to fully leverage global information to enhance cooperation
129 among workers. To tackle the challenges of large action and observation spaces, we propose a novel
130 network architecture, as illustrated in Fig. 2. This architecture employs the BERT model [11] to
131 effectively extract the relationships between workers and orders using the self-attention mechanism.
132 Additionally, an improved QK-attention [12] is implemented to reduce the computational complexity
133 associated with the order dispatching task. Furthermore, we introduce an action decomposition
134 method that breaks down the choice probability of each action within the vast action space into
135 individual action probabilities for each worker selecting each order. Finally, to address the data
136 scarcity challenge in MARL, we propose a two-stage training method, as shown in Fig. 1. In the
137 first stage, we train the upstream layers of the network using the IDDQN approach, allowing them
138 to develop general feature extraction capabilities. Subsequently, we train the entire neural network
139 using centralized TD3 to realize better cooperation between workers.

### 3.2 Network Architecture

141 The proposed network structure is shown as Fig. 2, which constists of three parts: encoders (embed
142 the worker and order information to a common feature space), actor sub-network (a BERT to
143 extract the relationship between different workers and orders and a QK-Attention to generate the
144 utility/probability of each worker-order pair), and critic sub-network (two BERT taking output of
145 actor BERT as input and output the Q-value respectively).

### 3.2.1 Feature Extractors

At each time step, the network takes the entire state $S_t = [W_t, O_t]$ as input. We consider this as a combination of two sequences: $W_t$ and $O_t$. For each element $w_{i,t}$ and $o_{j,t}$, we employ two distinct encoders, referred to as the "Worker Encoder" and the "Order Encoder", to embed them separately into a feature space of the same dimension, allowing them to be input into a single BERT model.

Each worker state $w_{i,t}$ consists of two parts: an on-board order sequence and other non-sequence information. For the order sequence, a bi-directional LSTM [13] is utilized to extract its features. This approach effectively encodes variable-length sequences into a uniform dimensional feature space, addressing the curse of dimensionality associated with conventional MLP encoders, where the number of parameters increases with sequence length. For non-sequence information, an MLP is employed for feature extraction. Finally, the two features are combined into a primary feature $\tilde{w}_{i,t}$. For the orders to be assigned $o_{j,t}$, an MLP is also used to extract the feature $\tilde{o}_{j,t}$. Notably, the dimensions of $\tilde{w}_{i,t}$ and $\tilde{o}_{j,t}$ are identical, and their information is concatenated into a sequence represented as $\tilde{S}_t = [\tilde{w}_{1,t}, \tilde{w}_{2,t}, \ldots, \tilde{w}_{n,t}, \tilde{o}_{1,t}, \tilde{o}_{2,t}, \ldots, \tilde{o}_{m_t,t}]$.

Additionally, to facilitate network convergence and enhance the extraction of input features, we incorporate a normalization layer and an Adaptive Re-weighting Layer (ARL) [14]. Given that different parts of the input may have varying magnitudes, which can impede model training, the normalization layer effectively addresses this issue. Furthermore, since different parts of the input carry different levels of importance, we utilize the ARL to enable the model to learn these variations, represented as: $y = x \circ \Omega$, where $x$ denotes the input, $\Omega$ represents the weight vector, calculated by $\Omega = \text{MLP}(x)$, and $\circ$ indicates the element-wise product.

### 3.2.2 Actor Sub-Networks

The Actor sub-network consists of a BERT [11] model for feature extraction and a QK-attention module [12] for action decomposition and generation, which we will introduce in turn. In the feature extractors, we have already extracted the primary features from each worker and order state separately. To further explore the relationships between workers and orders, we utilize the BERT model, where the self-attention mechanism can effectively capture these relationships: $\overline{S}_t = [\overline{w}_{1,t}, \overline{w}_{2,t}, \ldots, \overline{w}_{n,t}, \overline{o}_{1,t}, \overline{o}_{2,t}, \ldots, \overline{o}_{m_t,t}] = \text{Actor-BERT}(\tilde{S}_t)$. Specifically, due to the permutation invariance of our input sequence, we omit the positional embedding in BERT, ensuring that the order in $S$ does not influence the encoding result. In contrast to conventional MARL methods like [5; 7], which encode each worker with its neighboring states to gain a broader perspective, our Actor-BERT directly aggregates global worker information, facilitating more effective cooperative dispatching between workers.

In conventional order dispatching tasks, the typical approach to address the dynamic action space (related to the number of orders) involves evaluating each worker-order pair separately and finding the optimal dispatching solution based on these evaluations. However, this approach has two significant shortcomings. First, it neglects the relationships between orders, which we address through the self-attention mechanism in BERT, capturing not only the relationships between workers but also between orders and between orders and workers. Second, evaluating each worker-order pair is time-consuming and resource-intensive: $\text{F}(\overline{w}_{i,t}, \overline{o}_{j,t}; \theta_F) \in \mathbb{R}^1$, where F is the network and $\theta_F$ represents its parameters. The complexity can be represented as $O(|\text{F}| \cdot n \cdot m_t)$, where $|\text{F}|$ denotes the complexity of the neural network. To mitigate this issue, we employ a QK-attention module [12], represented as:

$$\text{QK-Attention}(\overline{w}_{i,t}, \overline{o}_{j,t}) := \text{f}(\overline{w}_{i,t}; \theta_f) \cdot \text{g}(\overline{o}_{j,t}; \theta_g)^T \approx \text{F}(\overline{w}_{i,t}, \overline{o}_{j,t}; \theta_F), \tag{2}$$

where f and g are two smaller networks, and $\theta_f$ and $\theta_g$ are their parameters. The intuition behind QK-attention is to use two smaller networks to approximate a larger network, similar to the motivation behind LoRA [15]. In this way, the complexity of computing all worker-order pairs becomes $O(|\text{f}| \cdot n + |\text{g}| \cdot m_t + d \cdot n \cdot m_t)$, where $|\text{f}|$ and $|\text{g}|$ are the complexities of the two neural networks, $d$ is their output dimension, and $d \cdot n \cdot m_t$ is the complexity of matrix multiplication. Here, $d$ is very small, making $d \cdot n \cdot m_t$ much smaller than the neural network computation complexity, i.e., $d \cdot n \cdot m_t \ll |\text{f}| \approx |\text{g}| < |\text{F}|$. Thus, we have $O(|\text{f}| \cdot n + |\text{g}| \cdot m_t + d \cdot n \cdot m_t) < O(|\text{F}| \cdot (n + m_t)) < O(|\text{F}| \cdot n \cdot m_t)$, indicating that the QK-attention successfully transforms the multiplication complexity of evaluating each worker-order pair into addition complexity.

However, we observe a parameter redundancy issue in Equation 2, which can lead to potential instability during training. This redundancy arises because there are actually infinite solutions for $f$ and $g$, as $f' = \alpha f$ and $g' = \frac{g}{\alpha}$ is also a valid solution for any non-zero real vector $\alpha$. Inspired by Dueling DQN [16], we propose a positive normalization method:

$$\text{QK-Attention-Norm}(\overline{w}_{i,t}, \overline{o}_{j,t}) := \text{f}(\overline{w}_{i,t}; \theta_f) \cdot \frac{\text{Softplus}(\text{g}(\overline{o}_{j,t}; \theta_g))^T}{||\text{Softplus}(\text{g}(\overline{o}_{j,t}; \theta_g))||_2} \ . \tag{3}$$

This normalization ensures that the elements in $\frac{\text{Softplus}(\text{g}(\overline{o}_{j,t};\theta_g)^T)}{||\text{Softplus}(\text{g}(\overline{o}_{j,t};\theta_g)^T)||_2}$ are always non-negative, with an L2 norm of 1. This guarantees a unique solution. In our task, the output of the QK-attention is a matrix $M_t \in \mathbb{R}^{n,m_t}$, representing the utility of each worker choosing each order, which will be detailed in Section 3.3.2.

### 3.2.3 Critic Sub-Networks

The role of the critic is to evaluate the quality of actions, with the detailed action generation method introduced in Section 3.3.2. We first define an action function $\mathcal{A}$:

$$\mathcal{A}(w_{i,t}) = \begin{cases} (\overline{w}_{i,t}, \overline{o}_{j,t}) & \text{if order } j \text{ is assigned to worker } i \text{ at time } t \\ \emptyset & \text{if no order is assigned to worker } i \text{ at time } t \end{cases} \tag{4}$$

where $\overline{w}_{i,t}$ and $\overline{o}_{j,t}$ are the outputs of Actor-BERT, and $(\overline{w}_{i,t}, \overline{o}_{j,t})$ represents the combination of the two vectors into a single feature vector. We then construct a new sequence: $\dot{S}_t = [\mathcal{A}(w_{1,t}), \mathcal{A}(w_{2,t}), \ldots, \mathcal{A}(w_{i,t})]$. Another BERT network, referred to as "Critic-BERT", is used to further extract features from $\dot{S}_t$, represented as $\ddot{S}_t = \text{Critic-BERT}(\dot{S}_t)$. A self-attention mechanism and a linear layer (collectively named Critic-MLP) are then utilized to estimate the Q-value from $\ddot{S}_t$ (for detailed processing methods, refer to [17]). Furthermore, as TD3 [9] requires two critics, we employ two distinct Critic-BERT and Critic-MLP networks. These share the input features from Actor-BERT but process them separately.

### 3.3 Training Process

#### 3.3.1 Stage 1: Decentralized IDDQN Training

In this stage, we aim to first train the feature-extracting capacity of the worker encoder and order encoder using a substantial number of samples. To obtain sufficient samples, we view the dispatching problem as a multi-agent scenario, where at each time step, each agent can access its own record. We adopt the independent assumption that all agents share the same policy, allowing for the sharing of records between agents and leading to a large experience replay buffer.

Since our goal in this stage is not to train a powerful model but rather to enable the feature extractor to learn its general feature-extracting capabilities, we select the simplest yet efficient method for order dispatching, namely, the IDDQN. Each worker is treated as an independent agent with the state defined as $s_{i,t} = [w_{i,t}, O_t]$ at time $t$. We employ a neural network to estimate the Q-value at each step as $Q_{\pi_\Phi^Q}^{DQN}(s_{i,t}, a_{i,t})$, where $\Phi$ represents the network parameters and $\pi_\Phi^Q$ denotes the strategy.

To construct the network, we utilize QK-attention to process the outputs of the worker encoder and order encoders to estimate the Q-value for each worker-order pair, represented as $\text{QK-Attention-Norm}(\tilde{w}_{i,t}, \tilde{o}_{j,t})$ (denoted as $y_{i,j,t}$). Although the state space encompasses the entire order state from $o_{1,t}$ to $o_{m_t,t}$, we focus on a single order $o_{j,t}$ when computing the Q-value for choosing order $j$. This approach aligns with previous work such as [5; 18], as the entire order state can be excessively large for a simple network to learn (our Triple-BERT effectively addresses this issue) and many networks struggle to process variable dimensional inputs (with order amounts varying at each time step). Consequently, we can compute a Q-matrix $Y_t \in \mathbb{R}^{n,m_t}$, where the element in the $i$-th row and $j$-th column, $y_{i,j,t}$, represents the Q-value of assigning order $j$ to worker $i$ at time $t$. The core strategy of IDDQN is to maximize the global Q-value, expressed as $Q(S_t, A_t) = \sum_{i=1}^{n} Q(s_{i,t}, a_{i,t})$ at each time step. To achieve this, we construct a bipartite graph where each worker and order is represented as a node. An arbitrary worker $i$ and order $j$ are linked by an edge weighted by the Q-value of this worker selecting this order at the current time, i.e., $y_{i,j,t}$. We then utilize Integer

Linear Programming (ILP) to solve this maximizing bipartite matching problem. (To avoid assigning orders to unavailable workers—those at full capacity or on their way to pick up an assigned order—we set the Q-value of all actions for such workers in the Q-matrix $Y_t$ to $-\infty$.) A detailed construction of the problem is provided in Appendix B.1. For the training of IDDQN, it follows the same process of previous work [5]. Due to page limitation, we detailed it in Appendix D.1.

### 3.3.2 Stage 2: Centralized TD3 Training

In the standard AC framework, the process can be summarized as follows: an actor network generates actions based on the current state, represented as $A_t = \text{Actor}(S_t; \theta_A)$, while a critic network evaluates these actions using $\hat{Q}_t = \text{Critic}(S_t, A_t; \theta_C, \pi_{\theta_A}^T)$. Here, $\theta_A$ and $\theta_C$ are the parameters of the actor and critic networks, respectively, and $\pi_{\theta_A}^A$ denotes the strategy of AC. During training, the critic network is updated using TD-error, similar to Q-learning, and the actor network is updated to maximize $\hat{Q}$. However, a challenge mentioned in Section 2 is that the action space is too large for the order dispatching scenario. Additionally, the actions in order dispatching are discrete, complicating optimization using TD3. To address these issues, we propose an action decomposition method along with a policy gradient-style optimization method.

Before delving into the details, we denote both $\theta_A$ and $\theta_C$ with the parameters $\Theta$, as in our network (Fig. 2), the actor and critic share the same architecture. The trained network parameters from Stage 1, $\Phi$, are part of $\Theta$. Moreover, the policy of TD3 is represented as $\pi_\Theta^T$.

**(i) Actor:** To address the large action space, we propose an action decomposition method that separates the probability of selecting each worker-order assignment combination into the probabilities of each worker choosing their respective orders. First, we expand the utility matrix $M_t$ output by the Actor QK-Attention to $\mathcal{M}_t = [M_t, N_t] \in \mathbb{R}^{n, m_t+1}$, where $N_t$ is an $n$-dimensional vector representing the utility of each worker choosing no order. This vector can be obtained by processing the output of Actor-BERT with a MLP, i.e., $N_t = \text{MLP}([\overline{w}_{1,t}, \overline{w}_{2,t}, \ldots, \overline{w}_{n,t}])$. This allows us to compute the probability of each worker choosing each action using a logit model [19], *if the actions among workers are independent*, i.e., $\mathscr{P}_t = \text{Softmax}(\mathcal{M}_t, \text{dim}=-1)$. According to this independent assumption, the joint action probability can be expressed as: $\prod_{i,j \in \text{h}(A_t)} \mathscr{P}_{i,j,t}$, where h() is defined as the function $\text{h}(A_t) = \{(i,j)|a_{i,j,t} = 1\}$. Similar to stage 1, we set the probability of those unavailable workers choosing no order to 1 and all other actions to 0 in $\mathscr{P}_t$. *(Remark: We consider this independent assumption can be approximately realized after the network is well-trained, as BERT has already captured the relationships among workers, including their strategic interactions.)*

However, since an order cannot be assigned to different workers repeatedly, the actions among workers are actually not independent. Intuitively, if a worker is more willing to choose a particular action, this action should have a higher probability of being selected by this worker in the joint action. Based on this intuition, the action choosing probability can be defined as:

$$\pi_\Theta^T(A_t|S_t) = \text{z}\left( \prod_{i,j \in \text{h}(A_t)} \mathscr{P}_{i,j,t} \right), \tag{5}$$

where $\text{z}(\cdot)$ is an increasing function that also depends on the current state $S_t$ (which we omit for simplicity). This equation implies that if an action $A_t$ has a higher value of $\prod_{i,j \in \text{h}(A_t)} \mathscr{P}_{i,j,t}$, it will have a higher probability of being chosen.

However, defining and computing such a function $\text{z}(\cdot)$ is challenging due to the vast action space, complicating the sampling of an action from the strategy $\pi_\Theta^T(A_t|S_t)$. We define an efficient approach to address this. First, during inference, we can greedily select the action with the maximum probability, as this action should theoretically have the highest utility:

$$\arg\max_{A_t \in \psi(S_t)} \pi_\Theta^T(A_t|S_t) = \arg\max_{A_t \in \psi(S_t)} \text{z}\left( \prod_{i,j \in \text{h}(A_t)} \mathscr{P}_{i,j,t} \right) = \arg\max_{A_t \in \psi(S_t)} \sum_{i,j \in \text{h}(A_t)} \log \mathscr{P}_{i,j,t} , \tag{6}$$

where $\psi(S_t)$ is the set of all possible actions under the current state $S_t$. This holds because both $\text{z}(\cdot)$ and $\log(\cdot)$ are increasing functions. We can construct a bipartite graph similar to Stage 1, where each available worker and order is represented as a node, and the link between each worker $i$ and order $j$ at time $t$ is weighted by their log probability $\log \mathscr{P}_{i,j,t}$. By utilizing ILP, we can find the action $A_t$ that maximizes $\pi_\Theta^T(A_t|S_t)$. The bipartite graph construction process is detailed in Appendix B.2. During

training, we introduce random noise to the probability matrix $\mathscr{P}_t$ and the model selects actions using the same method as in Eq. 6. When the noise is sufficiently large, the policy degrades to a totally random policy, and when the noise is zero, the policy converges to a greedy strategy. Although we cannot directly express the function $z(\cdot)$, it must ensure that the function is a increasing function (since the noise is totally random). More details about the noise can be found at Appendix C.

Optimizing this probability using vanilla TD3 is challenging due to the variable action space and the gap between action probabilities and the selected action (the gradient cannot propagate through them). To address this, we employ an approximate policy gradient optimization method [10]:

$$\nabla_\Theta J(\Theta) \propto \mathbb{E}_{\pi_\Theta^T}\left[(Q_{\pi_\Theta^T}^{TD3}(S_t, A_t) - B)\nabla_\Theta \sum_{i,j \in \mathsf{h}(A_t)} \log \mathscr{P}_{i,j,t}\right], \tag{7}$$

where $J(\Theta)$ is the optimization objective (long-term cumulative reward), $B$ is a baseline independent of state (we simplify by setting it to 0), and $Q_{\pi_\Theta^T}^{TD3}(S_t, A_t)$ is the Q-value under the policy $\pi_\Theta^T$, which can be estimated by $Q_{\pi_\Theta^T,i}^{TD3}(S_t, A_t; \Theta)$ using our proposed network ($i = 1, 2$, as there are two estimated Q-values in TD3). Detailed derivations can be found in Appendix C. We then use gradient ascent to maximize $J(\Theta)$, thus the loss function for the actor can be expressed as $L_A = -\nabla J(\Theta)$.

**(ii) Critic:** For the critic, it can be updated in a manner similar to vanilla TD3, where the loss function can be expressed as:

$$L_C = \sum_{i=1,2} \mathbb{E}_{\pi_\Theta^T}\left[\mathcal{Q}_{\pi_{\Theta^-}^T}^{TD3}(S_{t+1}, R_{t+1}; \Theta^-) - Q_{\pi_\Theta^T,i}^{TD3}(S_t, A_t; \Theta)\right],$$
$$\mathcal{Q}_{\pi_{\Theta^-}^T}^{TD3}(S_{t+1}, R_{t+1}; \Theta^-) = R_{t+1} + \gamma \min_{i=1,2} Q_{\pi_{\Theta^-}^T,i}^{TD3}(S_{t+1}, \mathrm{Actor}(S_{t+1}; \Theta^-, \xi); \Theta^-), \tag{8}$$

where $\mathcal{Q}_{\pi_{\Theta^-}^T}^{TD3}$ is the learning target function, $\Theta^-$ represents the parameters of the target network, which updates more slowly than the policy network $\Theta$ to provide a stable target, and $\xi$ is a small random noise applied in the probability matrix $\mathscr{P}$. More details can be viewed in Appendix D.2.

## 4 Experiment

Table 1: Comparison of Different Ride Sharing Methods

| Method | DeepPool [1] | BMG-Q [5] | HIVES [7] | Enders et al. [20] | CEVD [21] | Triple-BERT |
|---|---|---|---|---|---|---|
| **Type** | Independent | | CTDE | | Centralized | |
| **RL Algorithm** | IDDQN [22] | IDDQN [22] | QMIX [23] | MASAC [24] | VD[1] [25] | TD3 [9] |
| **Multi-Agent** | ✓ | ✓ | ✓ | ✓ | ✓ | × |
| **Network Backbone** | MLP | GAT [26] | GRU [27] | MLP+Attention | MLP | BERT [11] |
| **Model Size** | **20K** | 117K | 16M | 118K | 23K | 16M |
| **GPU Occupation (GB)** | **3.97** | 4.28 | 6.01 | 8.19 | 21.45 | 8.03 |
| **Average Reward** ($10^3$) | 12.72 | 13.04 | 12.37 | 12.04 | 13.16 | **14.73** |

To validate the proposed method, we evaluate its performance in the ride sharing dispatching task using real-world yellow ride-hailing data from Manhattan, New York City[2] [28]. To illustrate the efficiency and superiority of our proposed Triple-BERT, we compare it with several previous ride sharing methods of different types, including Independent MARL, CTDE MARL, and Centralized MARL, as shown in Table 1. Detailed information regarding our experiment configuration, simulator setup, and a comprehensive description of the comparative experiment can be found in Appendix E.

As shown in Fig. 3, we first illustrate the training process of different models by evaluating their performance in the training scenario every 10 episodes. The six sub-figures depict the cumulative reward, the number of orders served, and the average delivery time, detour time, pickup time, and confirmation time for each order. Additionally, the Greedy method serves as a baseline, where orders

---

[1]The original VD is a CTDE method. However, the CEVD variant modifies it to a centralized version.
[2]https://www.nyc.gov/site/tlc/about/tlc-trip-record-data.page

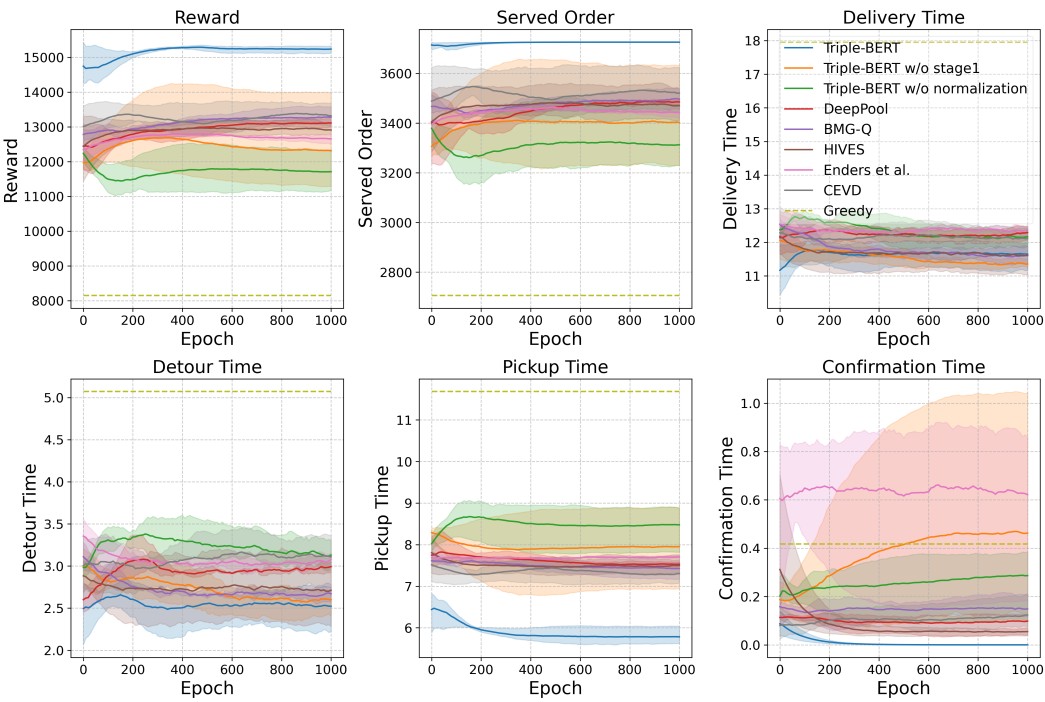

Figure 3: Training Process: Each method is trained three times, and the curve is smoothed using Exponential Moving Average (EMA) with $\alpha = 0.1$. The shaded area represents the range of fluctuations, while the solid line indicates the average value. (Here, for delivery time and detour time, only completed orders are counted, as these metrics are uncertain for unfinished orders.)

are assigned to the nearest worker. It is evident that our method outperforms the other models in most metrics, with the cumulative reward exceeding that of the best alternative method by approximately 15%. The highest number of served orders indicates that our method achieves better cooperation among workers. We then evaluate these methods over different periods, and the average rewards are shown in Table 1, where our method also demonstrates the best performance. More details about the experimental results can be found in Appendix E.4.

To further demonstrate the model's efficiency, we conduct a series of ablation studies. In terms of model training, we compare the performance of the model with and without stage 1 pre-training. Regarding the network structure, we primarily compare the QK-Attention mechanism with and without the proposed positive normalization module. The detailed results are shown in Fig. 3. We observe that without stage 1 pre-training, the model fails to converge and exhibits significant fluctuations. Particularly in the later stages, the reward begins to decrease, which can be attributed to the lack of samples. Additionally, without the proposed normalization in QK-Attention, the model performs poorly, underperforming compared to all other methods. This is due to parameter redundancy, which leads to substantial fluctuations and hinders efficient learning.

## 5 Conclusion

In this work, we propose the first centralized SARL method, Triple-BERT, for large-scale order dispatching in ride-hailing platforms. Our method successfully addresses the challenge of large action spaces through an action decomposition technique and tackles the issue of sample scarcity with a proposed two-stage training method. The novel network also addresses the large observation space challenge by leveraging the self-attention mechanism of BERT. Additionally, we introduce an improved QK-Attention mechanism to reduce the computational complexity of order dispatching. Through experiments on real-world ride sharing data, we demonstrate that our method significantly outperforms conventional MARL methods, achieving better cooperation among drivers.

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

# Appendix Contents

## A  Action Space Size

The action space in our order dispatching task is given by:

$$|A_t| = \sum_{k=0}^{m_t} \mathrm{C}(m_t, k)\mathcal{P}(n, k) = \sum_{k=0}^{m_t} \frac{m_t!}{k!(m_t - k)!} \frac{n!}{(n - k)!} \, , \tag{9}$$

where $\mathcal{P}(n, k)$ represents the permutations of assigning $k$ orders to $n$ workers and $\mathrm{C}(m_t, k)$ represents the combinations of selecting $k$ orders from the total $m_t$ orders. This equation is based on two assumptions: (i) the platform will assign an arbitrary number of orders at each step (some orders yielding negative income will be declined by the platform) and (ii) the number of orders $m_t$ is less than the number of workers $n$, which can always be satisfied since $m_t$ represents the order count at only one timestep. Then we can derive the lower bound of $|A_t|$ as:

$$\begin{aligned}
|A_t| &= \sum_{k=0}^{m_t} \mathrm{C}(m_t, k)\frac{n!}{(n - k)!} \geq \sum_{k=0}^{m_t} \mathrm{C}(m_t, k)(n - k + 1)^k \\
&\geq \sum_{k=0}^{m_t} \mathrm{C}(m_t, k)(n - m_t + 1)^k = (n - m_t + 2)^{m_t} \geq 2^{m_t} \quad (n \geq m_t \geq 0) \, .
\end{aligned} \tag{10}$$

As a result, the action space has a lower bound with the exponent to $m_t$. Consider the example in Section 2 where the number of workers $n$ is 1000 and the number of orders $m_t$ is 10. In this case, the expression $(n - m_t + 2)^{m_t}$ evaluates to $992^{10} \approx 10^{30}$.

## B  BiParite Graph Construction

### B.1  IDDQN Bipartite Graph

The bipartite graph in the IDDQN-based order dispatching method is constructed as follows:

$$\max_{A_t} \sum_{i \in \mathcal{I}} a_{i,j,t} \cdot y_{i,j,t}, \tag{11a}$$

$$\text{s.t.} \quad \sum_{i \in \mathcal{I}} a_{i,j,t} \leq 1, \quad \forall j \in \mathcal{J}_t, \tag{11b}$$

$$\sum_{j \in \mathcal{J}_t} a_{i,j,t} \leq 1, \quad \forall i \in \mathcal{I}, \tag{11c}$$

$$a_{i,j,t} \in \{0, 1\}, \quad \forall i \in \mathcal{I}, j \in \mathcal{J}_t, \tag{11d}$$

where $a_{i,j,t}$ is the action representing whether worker $i$ is assigned order $j$ at time $t$ (with 1 indicating assignment and 0 indicating no assignment), $y_{i,j,t}$ denotes the Q-value of worker $i$ choosing order $j$ at time $t$ (with $y_{i,j,t} = -\infty$ for all unavailable workers at time $t$), $\mathcal{I}$ is defined as $\{1, 2, \ldots, n\}$, and the set $\mathcal{J}_t$ is defined as $\{1, 2, \ldots, m_t\}$. Constraint 11b ensures that an order can be assigned to at most one worker, while constraint 11c guarantees that each worker is assigned at most one order.

### B.2  TD3 Bipartite Graph

The bipartite graph in our proposed TD3-based order dispatching method is constructed as follows:

$$\max_{X_t} \sum_{i \in \mathcal{I}_t^w} x_{i,j,t} \cdot \log \mathscr{P}_{i,j,t}, \tag{12a}$$

$$\text{s.t.} \quad \sum_{i \in \mathcal{I}} x_{i,j,t} \leq 1, \quad \forall j \in \mathcal{J}_t, \tag{12b}$$

$$\sum_{j \in \mathcal{J}_t} x_{i,j,t} = 1, \quad \forall i \in \mathcal{I}_t^w, \tag{12c}$$

$$x_{i,j,t} \in \{0, 1\}, \quad \forall i \in \mathcal{I}, j \in \mathcal{J}_t \cup \{m_t + 1\}, \tag{12d}$$

where $\mathcal{I}_t^w$ represents the set of available workers at time $t$. Here, constraint 12b does not apply in the last column, as it represents declining all orders, an action that can be chosen by any worker. Constraint 12c requires each row to equal 1, ensuring that each worker must either take an order or reject all, without other choices. We can then convert $X_t$ to action $A_t$ as follows:

$$a_{i,t} = \begin{cases} x_{i,j,t} & \text{if } i \in \mathcal{I}_t^w \text{ and } x_{i,m_t+1,t} = 0 \\ \mathbf{0} & \text{otherwise} \end{cases} \tag{13}$$

## C  Policy Gradient Proof

According to the policy gradient theory [10], we have:

$$\begin{aligned}
&\nabla_\Theta J(\Theta) \\
&\propto \mathbb{E}_{\pi_\Theta^T} \left[ \left( Q_{\pi_\Theta^T}^{TD3}(S_t, A_t) - B \right) \nabla_\Theta \log \pi_\Theta^T(A_t|S_t) \right] \\
&= \mathbb{E}_{\pi_\Theta^T} \left[ \left( Q_{\pi_\Theta^T}^{TD3}(S_t, A_t) - B \right) \nabla_\Theta \log \mathrm{z} \left( \prod_{i,j \in \mathrm{h}(A_t)} \mathscr{P}_{i,j,t} \right) \right] \\
&= \mathbb{E}_{\pi_\Theta^T} \left[ \left( Q_{\pi_\Theta^T}^{TD3}(S_t, A_t) - B \right) \frac{d\mathrm{z}(\prod_{i,j \in \mathrm{h}(A_t)} \mathscr{P}_{i,j,t})}{d\prod_{i,j \in \mathrm{h}(A_t)} \mathscr{P}_{i,j,t}} \frac{\prod_{i,j \in \mathrm{h}(A_t)} \mathscr{P}_{i,j,t}}{\mathrm{z}(\prod_{i,j \in \mathrm{h}(A_t)} \mathscr{P}_{i,j,t})} \nabla_\Theta \log \prod_{i,j \in \mathrm{h}(A_t)} \mathscr{P}_{i,j,t} \right] \\
&= \mathbb{E}_{\pi_\Theta^T} \left[ \left( Q_{\pi_\Theta^T}^{TD3}(S_t, A_t) - B \right) \mathcal{E}_{\mathrm{z}(x),x}|_{x=\prod_{i,j \in \mathrm{h}(A_t)} \mathscr{P}_{i,j,t}} \nabla_\Theta \sum_{i,j \in \mathrm{h}(A_t)} \log \mathscr{P}_{i,j,t} \right],
\end{aligned} \tag{14}$$

where $\mathcal{E}$ denotes elasticity, which measures the sensitivity of one variable to changes in another, and is defined as:

$$\mathcal{E}_{y,x} = \frac{d\log y}{d\log x} = \frac{dy}{dx}\frac{x}{y} . \tag{15}$$

Since $\mathrm{z}(x)$ is an increasing function, the elasticity is always non-negative. Here, we assume that the elasticity of $\mathrm{z}(x)$ with respect to $x$ can be approximately viewed as a positive constant. Thus, we have: $\nabla_\Theta J(\Theta) \propto \mathbb{E}_{\pi_\Theta^T} \left[ \left( Q_{\pi_\Theta^T}^{TD3}(S_t, A_t) - B \right) \nabla_\Theta \sum_{i,j \in \mathrm{h}(A_t)} \log \mathscr{P}_{i,j,t} \right]$, corresponding to Eq. 7.

*We acknowledge that the elasticity may not be a positive constant in practice (this requires that $z(x)$ has the same form as $ax^b$ $(a, b > 0)$). However, we consider this a reasonable approximation; otherwise, optimizing the actor would not be feasible, as obtaining a closed-form solution for $z(x)$ is impossible. Additionally, the final form of the equation aligns with the intuition that if an action has a higher Q-value, we should increase its probability, whereas we should decrease its probability if the Q-value is lower. While this approach may impede the model's convergence to the optimal solution, experimental results demonstrate the effectiveness of this formula, showing that it significantly outperforms other MARL methods.*

As mentioned in Section 3.3.2, during training, we add random noise to $\mathscr{P}_t$ and then choose the action that maximizes $\sum_{i,j \in \mathrm{h}(A_t)} \log \mathscr{P}_{i,j,t}$. Currently, the mapping from $\sum_{i,j \in \mathrm{h}(A_t)} \log \mathscr{P}_{i,j,t}$ to the choosing probability $\pi_\Theta^T$ corresponds to $\mathrm{z}(\cdot)$. To further illustrate the robustness of our method, we compare the performance of our model using Gaussian noise, uniform noise, and binary symmetric channel (BSC) noise, where the noise follows a Bernoulli distribution and has been widely utilized in previous work [5; 18]. During training, we gradually reduce the noise to make the policy more deterministic. The experimental results are shown in Fig. 4, where we observe that, despite certain performance differences between the various types of noise, they all outperform conventional MARL methods. This suggests the efficiency and high robustness of our proposed method, indicating that the detailed expression of $\mathrm{z}(x)$ does not significantly influence the validation of the method based on Eq. 7, even if it may cause some performance gaps. The optimal noise for our task may require further exploration. *For fairness, we choose to use BSC noise when comparing with other methods, even though it appears to perform the worst among the three types of noise. We aim to demonstrate that our results are robust and superior, not relying on a particular choice of hyper-parameters or experiment scenarios.*

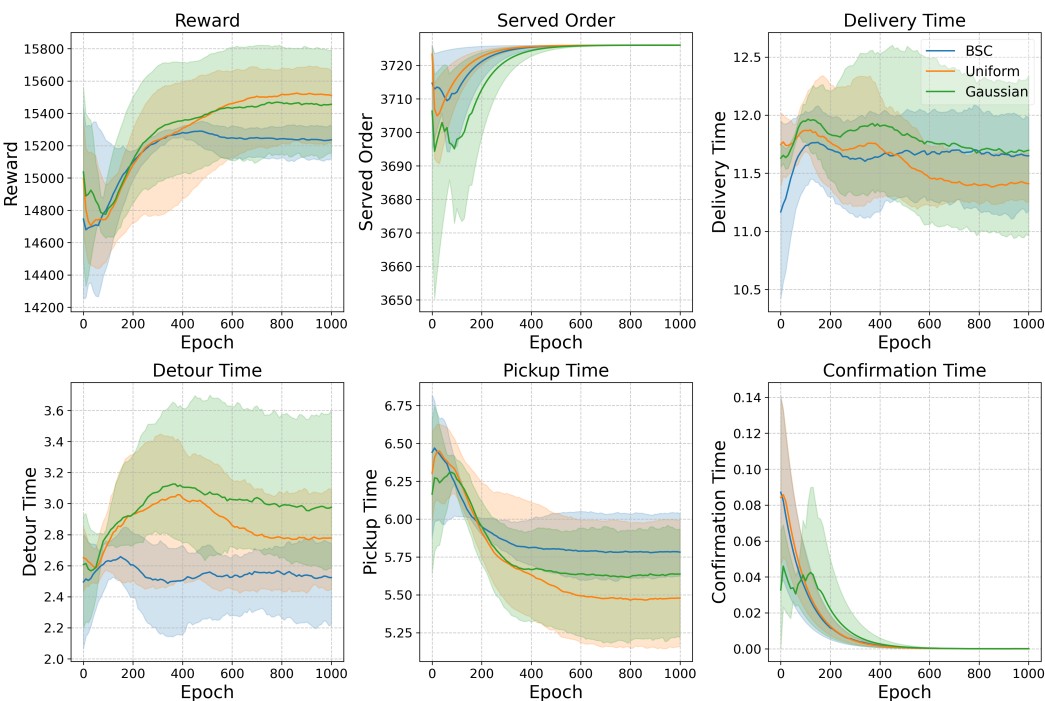

Figure 4: Comparison Between Different Noise Methods

## D Training Process

### D.1 Stage 1: IDDQN Algorithm

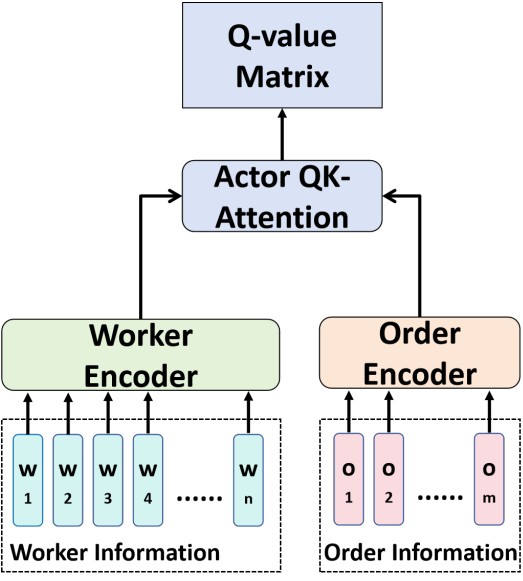

Figure 5: Network Structure in Stage 1

In stage 1, the network structure is shown as Fig. 5, which is consisted by the encoders and the QK-Attention module of proposed network in Fig. 2. *Remark: Although the model takes the entire worker and order sequence as input, it primarily aims to utilize parallel computation to enhance computational efficiency. In the encoders, each worker and order's information is processed separately. Similarly, in the QK-Attention module, the Q-value for each worker-order pair is computed*

504 *independently. It is also feasible to input only a single worker-order pair into this network, computing*
505 *the Q-value exclusively for that pair; however, this would increase the computation time.*

506 During IDDQN training, we need to introduce some noise into the Q-matrix $Y_t$ to facilitate sufficient
507 exploration. Specifically, for the $\epsilon$-greedy strategy, we randomly select a proportion $\epsilon$ of non-$-\infty$
508 elements in $Y_t$ and set them to a large positive number $\overline{Y}$ to enhance their likelihood of being selected.
509 We then update the neural network by minimizing the TD-error, expressed as:

$$
\begin{aligned}
L_Q &= \mathbb{E}_{\pi_\Phi^Q} \left[ \mathcal{Q}_{\pi_{\Phi^-}^Q}^{DQN}(s_{i,t+1}, r_{i,t+1}; \Phi^-) - Q_{\pi_\Phi^Q}^{DQN}(s_{i,t}, a_{i,t}; \Phi) \right] , \\
\mathcal{Q}_{\pi_{\Phi^-}^Q}^{DQN}(s_{i,t+1}, r_{i,t+1}; \Phi^-) &= r_{i,t+1} + \gamma Q_{\pi_{\Phi^-}^Q}^{DQN}(s_{i,t+1}, \kappa_{i,t+1}; \Phi^-) , \\
\kappa_{i,t+1} &= \arg \max_{\kappa_{i,t+1} \in \psi_{i,t+1}} Q_{\pi_\Phi^Q}^{DQN}(s_{i,t+1}, \kappa_{i,t+1}; \Phi) ,
\end{aligned}
\tag{16}
$$

510 where $\mathcal{Q}_{\pi_{\Phi^-}^Q}^{DQN}$ is the learning target function, $\gamma$ is the discount factor, $\psi_{i,t+1}$ is the possible action
511 space for worker $i$ at time $t+1$, and $\Phi^-$ represents the parameters of the target network, which are
512 updated at a slower pace compared to the policy network to provide a stable target for training. After
513 each training iteration, the target network is updated in a soft manner: $\Phi^- := \tau\Phi + (1-\tau)\Phi^-$,
514 where $\tau$ is the update rate.

515 The detailed process is illustrated in Algorithm 1, where $\mathbf{1}_j$ represents the vector that only the $j^{th}$
516 position is 1 and other positions are 0.

---

**Algorithm 1** IDDQN Training Process

---

**Require:** Number of training episodes $E$, number of training steps $T$, mini-batch size $m$, target
update rate $\tau$, exploration noise $\epsilon$, final exploration $\epsilon_f$, exploration decay $\delta$, discount factor $\gamma$,
model parameters $\Phi$
1: Initialize target networks $\Phi^- \leftarrow \Phi$
2: Initialize replay buffer $\mathcal{B}$
3: **for** $k = 1$ to $E$ **do**
4:     **for** $t = 1$ to $T$ **do**
5:         Calculate Q-value matrix $Y_t$: $y_{i,j,t} = Q_{\pi_\Phi^Q}^{DQN}(s_{i,t}, \mathbf{1}_j; \Phi)$
6:         Select action with exploration noise:$A_t = \text{ILP}(Y_t, \epsilon)$
7:         Observe reward $r_{i,t+1}$ and new state $s_{i,t+1}$ for each worker $i$
8:         Store transition $(s_{i,t}, a_{i,t}, r_{i,t+1}, s_{i,t+1})$ in $\mathcal{B}$
9:         Sample mini-batch of $m$ transitions $(s, a, r, s')$ from $\mathcal{B}$
10:        Compute target Q-value:
11:        $y \leftarrow r + \gamma Q_{\pi_{\Phi^-}^Q}^{DQN}(s_{i,t+1}, \arg\max_{\kappa_{i,t+1} \in \psi_{i,t+1}} Q_{\pi_\Phi^Q}^{DQN}(s_{i,t+1}, \kappa_{i,t+1}; \Phi); \Phi^-)$
12:        Update critics: $\Phi \leftarrow \arg\min_\Phi \frac{1}{m} \sum (y - Q_{\pi_\Phi^Q}^{DQN}(s, a; \Phi))^2$
13:        Update target networks: $\Phi^- \leftarrow \tau\Phi + (1-\tau)\Phi^-$
14:     **end for**
15:     Decay exploration: $\epsilon \leftarrow \max(\epsilon_f, \epsilon\delta)$
16: **end for**

---

### D.2 Stage 2: TD3 Algorithm

518 The process of our Stage 2 - TD3 training is illustrated in Algorithm 2. In experiment, we follow the
519 vanilla TD3 approach of updating the actor once after updating the critic twice.

**Algorithm 2** TD3 Training Process

---

**Require:** Number of training episodes $E$, number of training steps $T$, mini-batch size $m$, policy delay $d$, target update rate $\tau$, exploration noise $\epsilon$, final exploration $\epsilon_f$, exploration decay $\delta$, target policy smoothing noise $\xi$, discount factor $\gamma$, model parameters $\Theta$

1: Initialize target networks $\Theta^- \leftarrow \Theta$
2: Initialize replay buffer $\mathcal{B}$
3: **for** $k = 1$ to $E$ **do**
4:     **for** $t = 1$ to $T$ **do**
5:         Select action with exploration noise:$A_t = \text{Actor}(S_t; \Theta, \epsilon)$
6:         Observe reward $R_{t+1}$ and new state $S_{t+1}$
7:         Store transition $(S_t, A_t, R_{t+1}, S_{t+1})$ in $\mathcal{B}$
8:         Sample mini-batch of $N$ transitions $(S, A, R, S')$ from $\mathcal{B}$
9:         Compute target action with smoothing noise: $A' \leftarrow \text{Actor}(S; \Theta^-, \xi)$
10:        Compute target Q-value: $y \leftarrow r + \gamma \min_{i=1,2} Q^{TD3}_{\pi^T_{\Theta^-},i}(S', A'; \Theta^-)$
11:        Update critics: $\Theta \leftarrow \arg\min_\Theta \frac{1}{m} \sum [(y - Q^{TD3}_{\pi^T_\Theta,1}(S, A; \Theta))^2 + (y - Q^{TD3}_{\pi^T_\Theta,2}(S, A; \Theta))^2]$
12:        **if** $t \mod d == 0$ **then**
13:            Update actor using deterministic policy gradient:
14:            $\nabla J(\Theta) = \frac{1}{m} \sum (Q^{TD3}_{\pi^T_\Theta,1}(S, A; \Theta) - B) \nabla_\Theta \log \pi^T_\Theta(A_t|S_t), \quad (A = \text{Actor}(S; \Theta))$
15:            Update target networks: $\Theta^- \leftarrow \tau\Theta + (1 - \tau)\Theta^-$
16:        **end if**
17:     **end for**
18:     Decay exploration: $\epsilon \leftarrow \max(\epsilon_f, \epsilon\delta)$
19: **end for**

---

# E   Experiment Details

## E.1   Experiment Configurations

Our model was trained using the PyTorch framework [29] on a workstation running Windows 11, equipped with an Intel(R) Core(TM) i7-14700KF processor and an NVIDIA RTX 4080 graphics card. The detailed model configurations are shown as Table 2. During the training phase, the model utilized approximately 8.03 GB of GPU memory. For optimization, we employed the Adam optimizer with an initial learning rate of $10^{-4}$ and a decay rate of 0.99. In Stage 1, the batch size was set to 256, while in Stage 2, it was reduced to 16, due to a sharp decrease in sample amount. Additionally, optimization was performed once every 4 time steps, and in Stage 2, the actor was updated once for every two updates of the critic.

Table 2: Model Configurations

| Configuration | Our Setting |
| --- | --- |
| Hidden Dimension | 64 (Actor) / 128 (Critic) |
| Attention Heads | 4 |
| BERT Layers | 3 for Each |
| Dropout Rate | 0.1 |
| Optimizer | Adam |
| Learning Rate | $10^{-4}$ |
| Scheduler | ExponentialLR |
| Learning Rate Decay | 0.99 |
| Batch Size | 256 (Stage 1) / 16 (Stage 2) |
| Exploration Rate | $0.99 \rightarrow 0.0005$ |
| Updating Rate of Target Network | 0.005 |
| Discount Factor | 0.99 |

## E.2 Simulation Setup

In the simulation, we set the total number of drivers to 1,000, with each car having a capacity of 3 passengers. Each episode lasts 30 minutes, divided into 30 time steps, where each step determines the operations for the subsequent minute. For the TSP route optimization and time estimation, we utilize the OSRM simulator [30], with a default traveling speed of 60 km/h. We train the model using data from 19:00 to 19:30 on July 17, 2024, which includes 3,726 valid orders, and we test the trained model during other time periods on July 17, 2024, including 14:00-14:30 (2,850 valid orders), 17:00-17:30 (3,577 valid orders), 20:00-20:30 (3,114 valid orders), 21:00-21:30 (4,264 valid orders), and 22:00-22:30 (4,910 valid orders), where the order amount range from 2,850 to 4,264.

## E.3 Introduction of Comparative Methods

The methods using in our comparative experiment can be mainly divided into three categories:

- **Independent MARL:** The DeepPool [1] and BMG-Q [5] utilize a similar IDDQN method as described in Section 3.3.1, with BMG-Q employing GAT [26] to capture the relationships among neighboring agents. Additionally,in the original paper fo DeepPool, the authors used CNN. However, due to differences in the observation space of our task, we replaced it with MLP.
- **Centralized Training Decentralized Execution (CTDE):** The HIVES [23] framework introduces a QMIX [23] based method to address the shortcomings of IDDQN, specifically the inadequacy of treating the global Q-value as a simple summation of the individual Q-values of each agent. Enders et al. [20] propose a MASAC [24] based approach, allowing each driver to choose whether to accept an order, thereby preventing low-profit orders from negatively impacting the global income.
- **Centralized Training and Centralized Execution (CTCE):** CEVD [21], based on VD [25], innovatively combines the Q-values of each agent with those of their neighbors to create a new type of Q-value, akin to the motivation behind BMG-Q.

Overall, most of these methods attempt various strategies to enhance each agent's awareness of the global state, facilitating better cooperation. In contrast, our method directly transforms the formulation into a centralized single-agent reinforcement learning approach.

It is noteworthy that these Independent and CTDE MARL dispatching methods differ slightly from general MARL methods. In order dispatching, one order cannot be assigned to multiple workers, making it necessary to employ some centralized mechanism to achieve this. We refer to them as independent MARL and CTDE methods because they can directly calculate their own Q-values or action probabilities using their own or neighboring states. Conversely, CEVD must calculate the primary Q-value of each agent separately and then combine those primary Q-values with their neighbors to obtain a final Q-value for each agent.

Through the experimental results in Fig. 3, we observe that DeepPool [1], serving as one of the earliest benchmarks, demonstrates relatively stable and good performance, suggesting the simplicity and effectiveness of IDDQN features. In contrast, BMG-Q [5] significantly improves performance by utilizing FAT to capture neighboring information. As for HIVES [23] and CEVD [21], while they exhibit relatively good performance in the early stages of training—likely due to their hierarchical structure and centralized training methods—their performance becomes unstable in later stages, with rewards even starting to decrease. This instability may stem from the hierarchical approach not adequately addressing the large network input of the mixture network in QMIX and the lazy agent problem in VD. Additionally, their centralized training approach faces the same data scarcity issues as our method, making convergence more challenging. For Enders et al. [20], we note that their method shows worse performance than others. This may be related to their state processing method during training, where they replace the next state in the replay buffer with the request state from the current state to maintain a consistent agent count across two successive time steps, which appears to be a strong assumption. Finally, for the last three methods, their original papers primarily focus their reward functions on the serving order amount, without incorporating additional terms like ours (which also considers income, outcome, and user satisfaction levels). This makes our scenario more complex and may further reduce the performance of their methods in our setting.

 **E.4   Additional Experiment Result**

The detailed experimental results across different time periods are shown in Fig. 6, while the weighted average numerical results are presented in Table 3. For each model in each scenario, we repeat the experiment three times, and the error bars in the figure represent the standard deviation. We observe that our Triple-BERT achieves the highest reward across all scenarios, with the advantage becoming more pronounced as the order volume increases. Triple-BERT primarily optimizes the service rate and pickup time, significantly outperforming other methods.

For delivery time and detour time, the figures only account for completed orders, as the status of unfinished orders is uncertain, which may introduce some bias in the detailed values. In terms of these two metrics, Triple-BERT clearly performs better in high order volume scenarios, but not in low order volume scenarios. This may be due to the relatively low conflict caused by MARL in low order scenarios, while in high order scenarios, both the observation and action spaces increase sharply, making it challenging for MARL to find optimal solutions.

Lastly, we note that our method and the approach by Enders et al. [20] exhibit higher confirmation times. This may be attributed to both methods having an explicit rejection action (i.e., choosing no order), unlike the other methods. While this mechanism can lead to higher confirmation times, it also enables the model to discard negative profit orders and reserve some orders for currently unavailable workers.

Table 3: Average Performance under Multiple Periods

| Method | Reward | Service Rate | Delivery Time | Detour Time | Pickup Time | Confirmation Time |
|---|---|---|---|---|---|---|
| **DeepPool** [1] | 12723.85 | 0.91 | 11.53 | 2.47 | 7.77 | 0.06 |
| **BMG-Q** [5] | 13036.29 | 0.92 | **10.57** | **1.90** | 7.61 | 0.10 |
| **HIVES** [7] | 12365.11 | 0.89 | 11.04 | 2.28 | 7.99 | **0.03** |
| **Enders et al.** [20] | 12041.62 | 0.90 | 12.28 | 2.90 | 7.94 | 0.80 |
| **CEVD** [21] | 13157.96 | 0.94 | 11.36 | 2.31 | 7.37 | 0.06 |
| **Triple-BERT** | **14730.48** | **0.98** | 11.53 | 2.52 | **5.73** | 0.13 |
| **w/o stage 1** | 10665.02 | 0.87 | 11.92 | 2.72 | 9.36 | 0.68 |
| **w/o normalization** | 10839.33 | 0.88 | 12.50 | 2.85 | 9.10 | 0.24 |

# F   Discussions

## F.1   Limitations and Future Works

The limitations of this paper can be mainly categorized into two parts.

First, regarding the theoretical aspect, the current policy gradient formula is an approximation where we assume that the probability mapping function $z(x)$ has a nearly constant elasticity with respect to the independent variable $x$. Since obtaining a closed-form solution or elasticity for $z(x)$ is impossible, we must make certain assumptions for optimization. Although we have demonstrated the efficiency of Eq. 7 through intuition and experiments, there may still be a gap between the model's performance and the optimal solution. In future work, it would be valuable to explore an action strategy that can be proven to have elasticity to $z(x)$ close to a constant.

Second, concerning the experimental aspect, due to limitations of the experimental setup, we currently train and evaluate the model within a 30-minute simulation window. For 1,000 episodes, we can collect only 30,000 samples in a single-agent setting, which takes about a whole day to train a single method. This is why we designed the two-stage training method; otherwise, the model would struggle to converge with the limited samples. Future exploration should address whether stage 1 training is still necessary when the sample size increases. We also intend to investigate the model's performance in more diverse transportation scenarios, such as food delivery.

Finally, to better align with practical application scenarios and conditions, we plan to further develop the method to jointly optimize repositioning, payment, and price-setting tasks, making it more feasible for real-world use.

### F.2 Societal Impacts

This work has potential value for both academic research and practical applications in the transportation field, particularly for large-scale order dispatching tasks. By shifting from the conventional MARL paradigm to a SARL approach, we significantly improve model performance. This technology holds promise for enhancing daily travel and logistics transport.

However, the issue of algorithmic discrimination has received widespread attention over time. Closed-box management algorithms, including those for order dispatching, have been shown to create discriminatory scenarios for workers, as reinforcement learning methods primarily aim to maximize rewards without considering ethical implications. For example, algorithms may set different payment structures or order assignment preferences based on individual features or geographical locations of workers.

We hope that our method will not exacerbate these issues and can be further developed to include constraints that promote fairness. Our goal is to strike a balance between profit and ethics, fostering a win-win situation for platforms, workers, and customers.

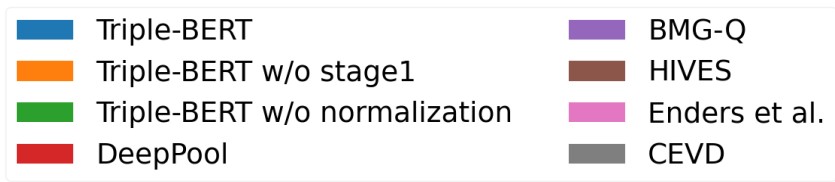

(a) Legend

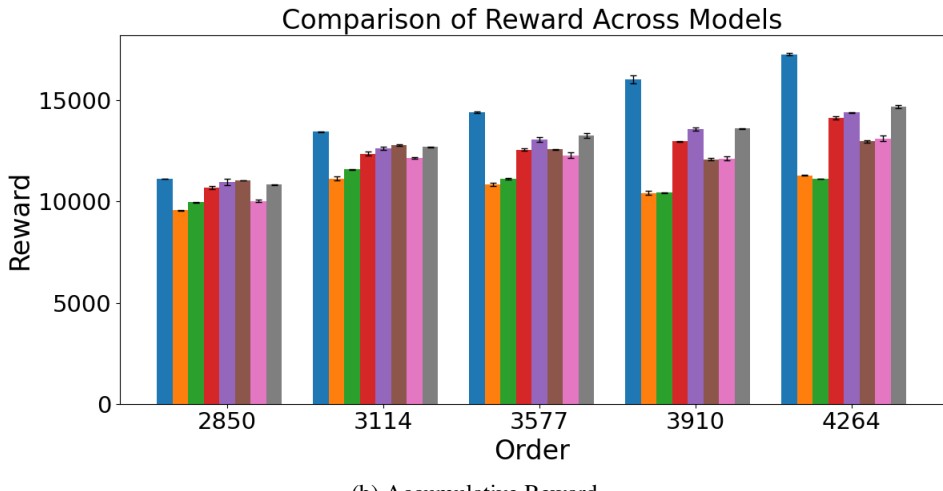

(b) Accumulative Reward

Figure 6: Detailed Evaluation Results

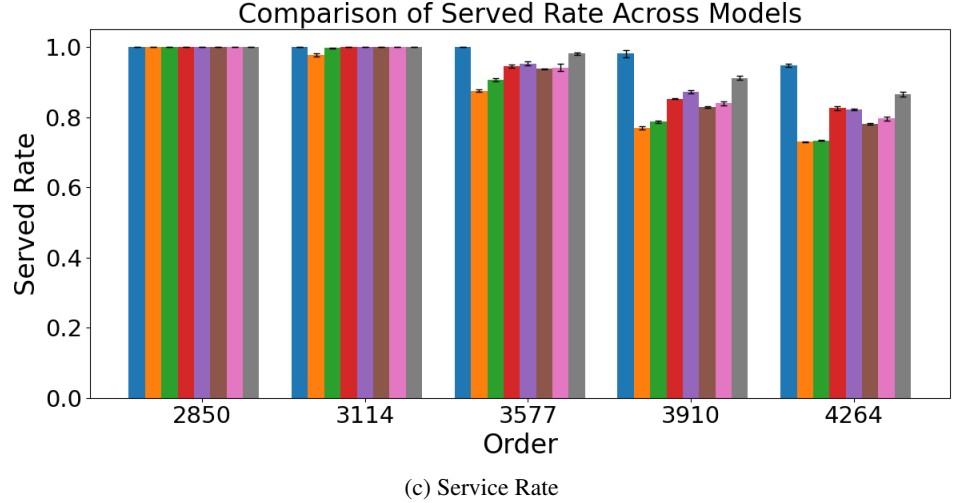

(c) Service Rate

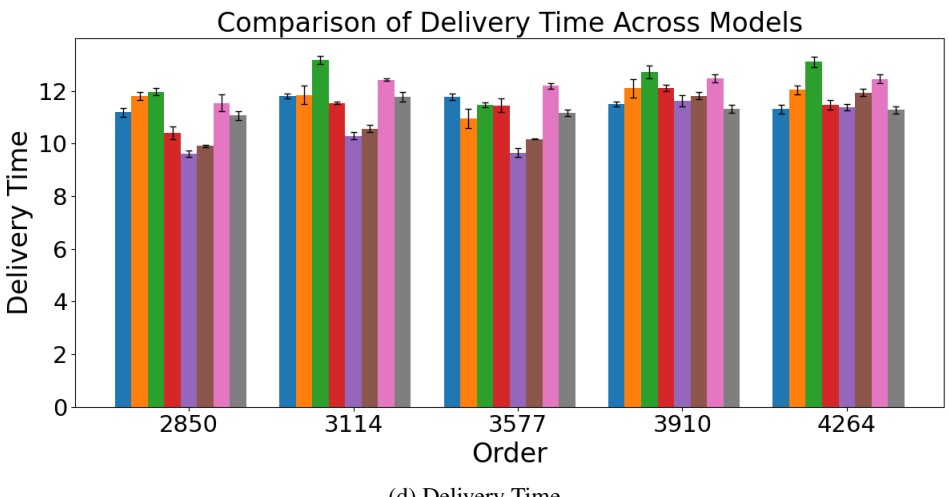

(d) Delivery Time

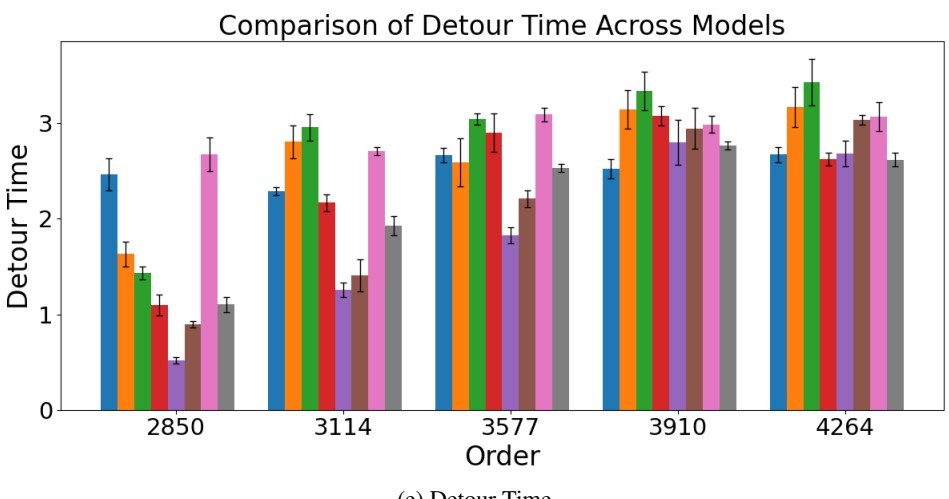

(e) Detour Time

Figure 6: Detailed Evaluation Results

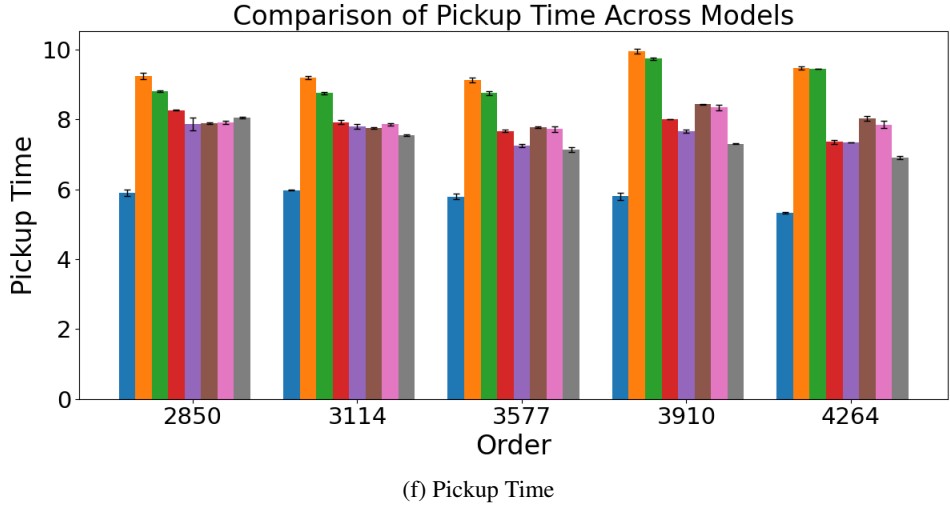

(f) Pickup Time

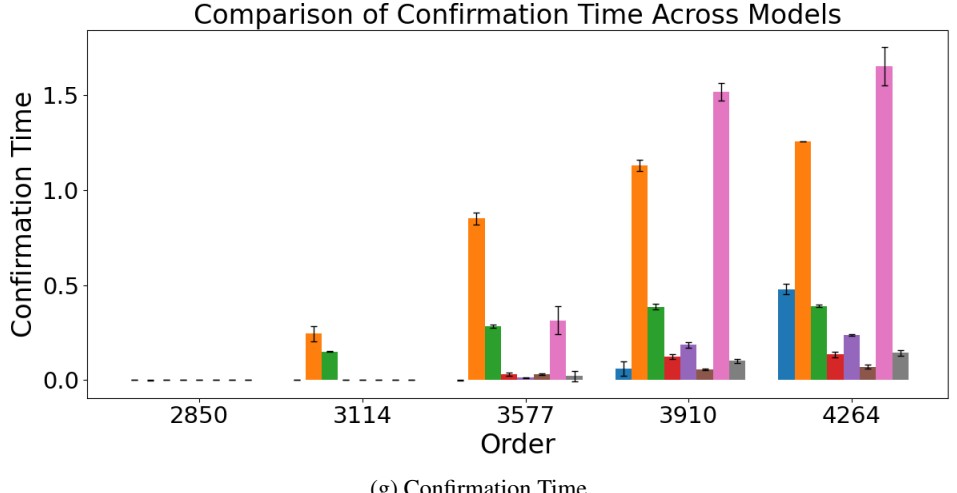

(g) Confirmation Time

Figure 6: Detailed Evaluation Results

