# OpenReview forum: "Triple-BERT: Do We Really Need MARL for Order Dispatch on Ride-Sharing Platforms?"
_NeurIPS.cc/2025/Conference — Submitted to NeurIPS 2025_

### Official Review · Reviewer_9Qmh · 2025-06-30

**Clarity:** 2
**Significance:** 2
**Originality:** 2
**Rating:** 3
**Confidence:** 3

**Summary:**

The authors consider large-scale real-time order matching under dynamic constraints (location, capacity, time windows). The authors propose a centralized Single-Agent RL (SARL) framework, where the input is the global state (driver pool + order pool), and the output is the driver-order assignment via action decomposition.

**Questions:**

More ablation studies could be helpful in the future version.
(1) The impact of the attention-based encoder on the experimental results;
(2) The impact of the attention-based encoder when applied to other algorithms/baselines;
(3) Whether adding/removing positional embeddings in transformer layers has negligible effects on results, thereby validating state permutation invariance?

**Ethical Concerns:**

["NO or VERY MINOR ethics concerns only"]

**Final Justification:**

After reading the response, my comment and scores remain the same.

**Limitations:**

NA.

**Paper Formatting Concerns:**

NO.

**Quality:**

2

**Strengths And Weaknesses:**

Strengths:
1. Performance and Scalability: The proposed SARL algorithm indeed outperforms other MARL algorithms, including DTDE, CTDE, and centralized methods.
2. Large-scale experiments: The algorithm effectively handles the dimension curse, involving 1000 agents with limited GPU memory, which has real-time deployment potential.

Weaknesses:
1. Independent assumption: Both the integration of IDDQN pre-training into the global value function and the decomposition of joint action into every driver rely on an independent assumption. However, the assumption lacks evidence, explanation, or verification.
2. Transformer encoder: the state input dimension is large, which makes the algorithm's generalization capability highly sensitive to the state encoder design.
3. Experiment design: The episode length is short: 30-minute episodes (30 steps), which ignores long-term coordination effects. Also, there is a lack of generalization validation across different days, rather than solely cross-validation over intra-day time periods.
4. It would be better to concisely and clearly illustrate the Stage 1 training process in Figure 2 or use another clear and illustrative figure to demonstrate the structure of IDDQN within the overall algorithm.
5. Typos: Eq. 5-6: Incorrect bracket sizing.

---

> ### Author Rebuttal · Authors · 2025-07-30
>
> We are grateful for your thoughtful feedback. Your assessment is valuable for us to polish our paper. We are glad to address your questions.
>
> > W1: Independent Assumption Issue
>
> **Re:** Thank you for your insightful comment. We acknowledge that the independent assumption can be unreliable, which indeed hinders the performance of conventional independent MARL-based ride-sharing methods [1-2]. A key contribution of our paper is addressing this limitation by employing a centralized SARL method, which considers the states and actions of all drivers collectively, thereby facilitating improved cooperation.
>
> In our approach, the independent assumption is only utilized during the pre-training stage to warm up the model and provide a solid starting point for centralized training. After pre-training, our centralized learning framework no longer relies on the independence assumption; thus, it is used solely as an intermediate step to warm up the computation of the centralized framework, rather than a requirement of the overall approach. Our experimental results demonstrate that this initial guess is sufficient for our purposes.
>
> > W2 & Q1: Encoder Design Issue
>
> **Re:** Thank you for your insightful comment. In the paper, we have demonstrated the generalization capacity of our methods in Appendix E4 and Figure 6, where Triple-BERT outperforms other methods across all scenarios with varying order amounts, particularly in scenarios of high order volumes. To further illustrate the impact of the encoder design on generalization capability, we conducted a comparative study, where we replaced the current encoder with a simple three-layer MLP.
>
> **Table 1.** Reward of Triple-BERT (with and without our proposed attention-based encoder) in different scenarios
>
> | Order Amount     | w/ attention-based encoder | w/o attention-based encoder | previous  SOTA |
> | ---------------- | -------------------------- | --------------------------- | -------------- |
> | 3,726 (training) | **15,388** (+2.81%)        | 14,967                      | 13,682         |
> | 2,850            | **11,148** (+0.18%)        | 11,127                      | 11,044         |
> | 3,114            | **13,483**(+0.63%)         | 13,399                      | 12,787         |
> | 3,577            | **14,477**(+0.04%)         | 14,471                      | 13,064         |
> | 3,910            | **16,335**(+2.67%)         | 15,909                      | 13,609         |
> | 4,264            | **17,366**(+2.89%)         | 16,877                      | 14,637         |
>
> We observed that the attention-based encoder primarily improves performance in scenarios of high order volumes, with limited influence in scenarios of low order volumes. Additionally, even with the simple MLP-based encoder, Triple-BERT still outperforms previous MARL methods in all scenarios, illustrating the robustness of our centralized SARL method.
>
> > W3: Experiment Design Issue
>
> **Re:** Thank you for your insightful comment. We acknowledge that the short simulation time is a limitation of this paper, primarily due to resource constraints for training. This aspect can be left in future research. However, we believe that this duration is sufficient to demonstrate the superiority of our method over others, as similar 30-60 minute simulation time settings have been utilized in some comparative studies [1-3].
>
> In response to your suggestion regarding generalization, we compared our Triple-BERT with other methods using the testing scenario from July 16 to July 18, 2024, at 6 PM. The results indicate that our Triple-BERT achieved the highest reward among all scenarios, highlighting its strong generalization capacity.
>
> **Table 2.** Reward of Triple-BERT and previous MARL methods across different days
>
> | Testing Scenarios   | DeepPool | BMG-Q  | HIVES  | Enders et al. | CEVD   | Triple-BERT |
> | ------------------- | -------- | ------ | ------ | ------------- | ------ | ----------- |
> | 7.16 (4,451 orders) | 13,473   | 14,121 | 12,070 | 12,142        | 14,226 | **16,831**  |
> | 7.17 (4,125 orders) | 13,204   | 13,424 | 12,232 | 12,208        | 14,145 | **16,145**  |
> | 7.18 (3,635 orders) | 12,679   | 13,067 | 12,397 | 12,268        | 13,336 | **14,819**  |
>
> > W4: Figure Illustration Issue
>
> **Re:** Thank you for your insightful comment. We currently provide the network structure of IDDQN in Figure 5 in the appendix, along with the training process in Algorithm 1. Due to page limitations, it is challenging to further expand Figure 2. In the revised version, we plan to add a detailed training process workflow for the two-stage training in appendix, following the format of Figure 1 in [4]. This will clarify how we perform order assignment during exploitation and what the learning targets are during training.
>
> > W5: Typos
>
> **Re:** Thank you for pointing it out. We will correct it in the revised paper.
>
> > Q2: Encoder Issue of Baselines
>
> **Re:** Thank you for your insightful comment. We have compared the performance of other methods with and without our attention-based encoder to assess its impact on various algorithms and baselines.
>
> **Table 3.** Reward of previous MARL (with and without our proposed attention-based encoder) in training scenario
>
> | Method        | w/ attention-based encoder | w/o attention-based encoder |
> | ------------- | -------------------------- | --------------------------- |
> | DeepPool      | **14,570** (+9.29%)        | 13,332                      |
> | BMG-Q         | **13,879** (+2.51%)        | 13,539                      |
> | HIVES         | 13,095(-0.23%)             | **13,126**                  |
> | Enders et al. | **12,706** (+0.28%)        | 12,670                      |
> | CEVD          | 13,724 (-0.16%)            | 13,746                      |
>
> The results indicate that the attention-based encoder significantly enhances the performance of independent MARL methods such as DeepPool and BMG-Q. However, this improvement does not extend to CTDE and CTCE (centralized MARL) approaches like HIVES, Enders et al., and CEVD. The underlying reason is that CTDE and CTCE methods in ride-sharing primarily suffer from the curse of dimensionality in the critic network, an issue that cannot be mitigated by a more powerful feature extractor. We believe that the improvements observed in DeepPool, BMG-Q, and our Triple-BERT effectively enhance the efficiency of our designed encoder. Furthermore, even with the attention-based encoder, our Triple-BERT consistently outperforms all previous methods, underscoring the superiority of our approach.
>
> > Q3: Positional Embedding Issue
>
> **Re:** Thank you for your insightful comment. The reasons for eliminating the positional embeddings are as follows: (1) The localization information is already incorporated in the state of each worker and order, making external positional embeddings unnecessary. (2) Since there is no specific order relationship among workers and orders, determining a meaningful order is challenging. (3) When utilizing positional embeddings, a maximum length must be defined for the model. In practical applications, if the number of orders exceeds this maximum length at any point, the model will be unable to process them effectively. In light of your comment, we will compare the performance of the method with and without positional embeddings to further validate state permutation invariance.
>
> **Table 4.** Reward of Triple-BERT (with and without positional embedding) in different scenarios
>
> | Order Amount     | w/o positional embedding | w/ positional embedding |
> | ---------------- | ------------------------ | ----------------------- |
> | 3,726 (training) | **15,388** (+8.84%)      | 14,092                  |
> | 2,850            | **11,148** (+4.39%)      | 10,679                  |
> | 3,114            | **13,483**(+17.95%)      | 11,431                  |
> | 3,577            | **14,477**(+12.74%)      | 12,841                  |
> | 3,910            | **16,335**               | ×                       |
> | 4,264            | **17,366**               | ×                       |
>
> The results indicate that positional embedding does not enhance model training. On the contrary, it introduces additional interference, resulting in lower performance. This is particularly evident in generalization problems: when using positional embeddings, Triple-BERT only outperforms previous MARL SOTA in training scenarios (please refer to Table 1), but not in testing scenarios. Additionally, when the order amount exceeds the training scenario, the maximum length restriction hinders the model's effectiveness.
>
> ------
>
> Finally, we express our heartfelt gratitude for your insightful questions and suggestions. They have been invaluable to our work and will be integral to our revised manuscript. **We hope our efforts will adequately address your concerns and that you will recognize our work.**
>
> **References:**
>
> [1] Al-Abbasi A O, Ghosh A, Aggarwal V. Deeppool: Distributed model-free algorithm for ride-sharing using deep reinforcement learning[J]. IEEE Transactions on Intelligent Transportation Systems, 2019, 20(12): 4714-4727.
>
> [2] Hu Y, Feng S, Li S. Bmg-q: Localized bipartite match graph attention q-learning for ride-pooling order dispatch[J]. arXiv preprint arXiv:2501.13448, 2025.
>
> [3] Enders T, Harrison J, Pavone M, et al. Hybrid multi-agent deep reinforcement learning for autonomous mobility on demand systems[C]//Learning for Dynamics and Control Conference. PMLR, 2023: 1284-1296.
>
> [4] Shah S, Lowalekar M, Varakantham P. Neural approximate dynamic programming for on-demand ride-pooling[C]//Proceedings of the AAAI conference on artificial intelligence. 2020, 34(01): 507-515.

---

### Official Review · Reviewer_qVKq · 2025-07-01

**Clarity:** 3
**Significance:** 3
**Originality:** 3
**Rating:** 4
**Confidence:** 4

**Summary:**

This paper proposes Triple-BERT, a centralized Single-Agent Reinforcement Learning (SARL) approach for large-scale order dispatching in ride-sharing platforms. The authors challenge the conventional use of Multi-Agent Reinforcement Learning (MARL) by introducing a centralized method that leverages a BERT-based architecture with self-attention mechanisms to handle the complex relationships between drivers and orders. The approach includes an action decomposition strategy to manage the vast action space and a two-stage training methodology to address sample scarcity. Experimental results on Manhattan taxi data show approximately 11.95% improvement over state-of-the-art MARL methods.

**Questions:**

Q1 Is the two-stage training always necessary, or could end-to-end training work with sufficient data? What determines the transition point between stages?
Q2 How does the method scale with larger numbers of drivers and orders? Have you tested beyond 1000 drivers and what are the computational bottlenecks?
Q3 Can you provide principled guidelines for choosing the noise type and schedule? The performance differences between noise types seem significant but arbitrary

**Ethical Concerns:**

["NO or VERY MINOR ethics concerns only"]

**Limitations:**

Yes

**Quality:**

3

**Strengths And Weaknesses:**

S1 The problem is practically important and the solution could have significant impact on ride-sharing platforms.
S2 The paper provides thorough comparisons with multiple baselines across different categories (Independent MARL, CTDE, Centralized) and includes proper ablation studies.
S3 The paper is generally well-written with good mathematical formulation and clear methodology description. Very Nice illustrations
S4 The paper presents a compelling argument for moving from MARL to centralized SARL for order dispatching, which is theoretically sound and addresses real limitations of existing approaches.

W1 Evaluation is restricted to 30-minute simulation windows due to computational constraints (Mentioned by the authors)
W2 The necessity of the two-stage training suggests potential scalability issues with larger datasets

---

> ### Author Rebuttal · Authors · 2025-07-30
>
> We are grateful for your thoughtful feedback. Your assessment is valuable for us to polish our paper. We are glad to address your questions one by one.
>
> > W1 Evaluation is restricted to 30-minute simulation windows due to computational constraints (Mentioned by the authors)
>
> **Re:** Thank you for your insightful comment. We acknowledge that the limited simulation time is a shortcoming of our experiment. However, we want to emphasize that the main contribution of this paper is to propose a novel and efficient paradigm for ride-sharing platforms, specifically the first centralized SARL-based method. Despite the shorter simulation time, the superior performance of our method compared to others demonstrates its efficacy. Additionally, we believe the experiment is valid and fair, as a 30 to 60-minute simulation time is also commonly used in our comparative works [1-3].
>
> > W2 The necessity of the two-stage training suggests potential scalability issues with larger datasets
> >
> > Q1 Is the two-stage training always necessary, or could end-to-end training work with sufficient data? What determines the transition point between stages?
>
> **Re:** Thank you for your insightful comment. We believe that the two-stage training strategy is not strictly necessary but is indeed helpful and efficient. In centralized SARL, the sample efficiency is significantly lower than that of MARL methods during the same time periods. Specifically, while the MARL method can gather samples equivalent to the number of drivers at each step, the SARL method can only obtain one sample. Assuming both methods require the same number of samples for convergence, training SARL from scratch would require *n* times more episodes than MARL, where *n* is the number of drivers.
>
> In our experiments, we observed that with two-stage training, the SARL method can converge in the same timeframe as the MARL method. Thus, if the total episodes required for MARL convergence is *m*, then training SARL from scratch would typically require *mn* episodes, whereas our two-stage method only requires *2m* episodes (with each stage needing *m* episodes). Additionally, future research could explore whether mechanisms like offline pre-training and online fine-tuning could further enhance our method.
>
> Regarding the transition point between stages, we currently opt to train the MARL method to convergence first, followed by training the SARL method. In our experiments, each stage requires 1,000 episodes.
>
> > Q2 How does the method scale with larger numbers of drivers and orders? Have you tested beyond 1000 drivers and what are the computational bottlenecks?
>
> **Re:** Thank you for your insightful comment. To identify the computational bottlenecks, we tested the computation time with driver amounts ranging from 1,000 to 2,000 and order amounts from 300 to 500 at single time step. (In our real-world dataset, we typically observe that the order amount does not exceed 200 at any single step.) The results indicate that the decision time remains consistently under 0.2 seconds across all scenarios. Additionally, the order and driver counts have minimal impact on computation time. This suggests that, for the current simulation, most of the processing time is allocated to the simulator's operations rather than to decision computation cost.
>
> **Table 1:** Decision time of different driver and order amounts ('s' represents seconds)
>
> | Driver Amount \ Order Amount | 300      | 400      | 500      |
> | ---------------------------- | -------- | -------- | -------- |
> | 1,000                        | 0.1801 s | 0.1839 s | 0.1870 s |
> | 1,500                        | 0.1806 s | 0.1820 s | 0.1830 s |
> | 2,000                        | 0.1789 s | 0.1829 s | 0.1809 s |
>
> In our experiments detailed in the appendix E4 (Fig. 6), we have already evaluated our model's performance with an order amount reaching 4,264, which is close to the peak in the dataset.
>
> To further evaluate the scalability of our method, we compared its performance against other approaches as the driver count increased to 1,500 and 2,000 during a high concurrency period. In this scenario, the order volume reached 6,775, which we synthesized by combining orders from two different periods.
>
> **Table 2:** Reward of different methods during high concurrency period among different driver amount
>
> | Driver Amount | DeepPool | BMG-Q  | HIVES  | Enders et al. | CEVD   | Triple-BERT (ours) |
> | ------------- | -------- | ------ | ------ | ------------- | ------ | ------------------ |
> | 1,500         | 21,090   | 22,333 | 19,587 | 19,546        | 23,092 | **27,458**         |
> | 2,000         | 25,316   | 25,650 | 25,713 | 25,394        | 26,207 | **28,273**         |
>
> **Table 3:** Simulation time of different methods during high concurrency period among different driver amount
>
> | Driver Amount | DeepPool | BMG-Q | HIVES | Enders et al. | CEVD | Triple-BERT (ours) |
> | ------------- | -------- | ----- | ----- | ------------- | ---- | ------------------ |
> | 1,500         | **67 s** | 71 s  | 79 s  | 109 s         | 76 s | 79 s               |
> | 2,000         | **78 s** | 81 s  | 80 s  | 107 s         | 81 s | 84 s               |
>
> The results indicate that our Triple-BERT model achieves the highest reward across various scenarios, without the need for retraining. Although the simulation time is slightly longer, it remains acceptable, as our method completes a 30-minute simulation in under 1.5 minutes, which also accounts for the simulator's running time.
>
> > Q3 Can you provide principled guidelines for choosing the noise type and schedule? The performance differences between noise types seem significant but arbitrary.
>
> **Re:** Thank you for your insightful comment. The choice of noise type remains an open question in the field of RL particularly in ride-sharing applications. Currently, most studies select a feasible noise type without thorough explanation; for example, [1] opted for BSC-style noise, while [4] mentioned action noise without specifying the type. In Appendix C (Fig. 4), we compared the influence of different popular noise types and found that uniform and Gaussian noise yield similar performance, while BSC performed the worst. To highlight the robustness of our method, we chose to compare it against the worst-performing noise (BSC), demonstrating that our approach can achieve SOTA performance without relying on specific noise designs.
>
> In practice, to maximize the performance of our method, we recommend using continuous noise types like Gaussian or uniform noise instead of discrete types like BSC. The BSC noise can lead to failures in certain scenarios. For instance, if there is one available driver and two orders with probabilities p1 and p2 (where p1≫p2), as training progresses, the noise diminishes and the agent tends to exploit rather than explore. With Gaussian or uniform noise, order 2 would be unlikely to be chosen even with a negative disturbance to p1 and a disturbance to p2, aligning with the learned strategy. However, with BSC noise, if p2 receives a disturbance (which can only be a value of 1, with no other disturbances), order 2 will definitely be chosen, leading to over-exploration. This explains the decreasing trend in rewards for the BSC-based method in Figure 4, unlike the Gaussian and uniform methods.
>
> ------
>
> Finally, we express our heartfelt gratitude for your insightful questions and suggestions. They have been invaluable to our work and will be integral to our revised manuscript. **We hope our efforts will adequately address your concerns and that you will recognize our work.**
>
>  **References:**
>
> [1] Hu Y, Feng S, Li S. Bmg-q: Localized bipartite match graph attention q-learning for ride-pooling order dispatch[J]. arXiv preprint arXiv:2501.13448, 2025.
>
> [2] Enders T, Harrison J, Pavone M, et al. Hybrid multi-agent deep reinforcement learning for autonomous mobility on demand systems[C]//Learning for Dynamics and Control Conference. PMLR, 2023: 1284-1296.
>
> [3] Al-Abbasi A O, Ghosh A, Aggarwal V. Deeppool: Distributed model-free algorithm for ride-sharing using deep reinforcement learning[J]. IEEE Transactions on Intelligent Transportation Systems, 2019, 20(12): 4714-4727.
>
> [4] Lei Z, Ukkusuri S V. Scalable reinforcement learning approaches for dynamic pricing in ride-hailing systems[J]. Transportation Research Part B: Methodological, 2023, 178: 102848.

---

### Official Review · Reviewer_4BHb · 2025-07-10

**Clarity:** 2
**Significance:** 2
**Originality:** 2
**Rating:** 4
**Confidence:** 4

**Summary:**

This paper proposes Triple-BERT, a centralized single-agent reinforcement learning (SARL) framework for large-scale order dispatching on ride-sharing platforms. The key innovation is transforming the traditionally multi-agent problem into a centralized approach using BERT-based architecture with QK-attention mechanism. The method employs a two-stage training strategy: pre-training feature extractors with MARL (IDDQN) followed by centralized TD3 fine-tuning. Experiments on NYC taxi data show good improvement over MARL baselines.

**Questions:**

- What is the research problem you really want to focus on? If it is ride-sharing systems. I suggest solely using terminology that reflects the ride-hailing domain, avoiding using food/package delivery. Further, in the problem setup, modeling, and experiment, you should consistently ground your work in the actual business constraints and evaluation metrics of the ride-hailing domain. The current setup is confused. Why do "workers decline orders" not the customers? The pickup time and service rate were never introduced in the problem setup. How did they come up in the experiment section?
- Please re-examine your algorithmic choices. Why didn't you use DQN or its variants? If your method is better, at least, you need to demonstrate it through computational experiments.
- What is the definition of z(·)? If your approach relies on the assumptions of it being constant elasticity, you need to be transparent about what these mean and whether they are theoretically defensible. How do you solve the ILP? In reality, order dispatching should be finished in seconds. If solving ILP is time-consuming, your method is hard to apply in practice. What is the full name of IDDQN?
- If you want to claim progress in ride-sharing dispatch, the current experimental evaluation is narrow and unconvincing.
   - Operations research methods are also strong in this domain. Why didn't you compare with them?
   - There exist several RL-based dispatch systems from major platforms (e.g., DiDi’s work in Qin et al. 2020 and Xu et al. 2018). Consider comparing against these industrial baselines.
- Consider providing a dedicated related works section, which reviews the paper solving similar problems and the progress of RL or MARL on solving large-scale combinatorial problems.
- Consider changing the title of this paper. Why do you use "Tripe"?(Is it because of TD3? Three BERTs? This isn't clear.). The question format doesn't directly state the contribution. The answer is implicitly "no", but the paper shows "it's complicated".

**Ethical Concerns:**

["NO or VERY MINOR ethics concerns only"]

**Final Justification:**

The authors have addressed some of my concerns regarding the problem setting, providing additional clarity and context. However, the theoretical contributions of the paper remain insufficiently developed to meet the standard for acceptance.

**Limitations:**

Yes.

**Quality:**

2

**Strengths And Weaknesses:**

(1) Strengths
- The problem has a clear motivation. It uses data to argue that existing MARL approaches struggle with coordination at scale (1000+ drivers), making a centralized approach worthwhile to explore.
- It designs a novel architecture. The combination of BERT self-attention with QK-attention for driver-order utility estimation is creative and addresses computational complexity effectively.
- Code, trained models, and processed data are made available. Implementation details are well-documented in the appendices.

(2) Weaknesses
-  The paper fundamentally misunderstands the realities of ride-sharing dispatch. The paper claims to address order dispatch in ride-sharing platforms like Uber or Lyft, but its problem formulation is out of touch with how these systems actually work. Here are some evidence:
   - The authors repeatedly use “delivery” terminology (“minimize delivery time,” “food delivery platform”) rather than language appropriate for ride-hailing (where the focus is on matching drivers to passengers, not package delivery).
   - Their model ignores the time and distance a driver must travel to pick up a passenger, which is called “deadheading” in the industry. In reality, reducing this “pickup time” is a core challenge in ride-sharing logistics.
   - There is no concrete modeling of customer impatience or cancellation beyond a superficial mention, despite this being a key driver of system performance in ride-hailing.
- It is doubtful whether the authors are deeply versed in reinforcement learning fundamentals for this class of problem. The paper introduces the term “SARL”—Single-Agent RL—which is not used in mainstream RL literature. In RL, a single-agent setting is just called RL; the SARL label is, at best, a distraction. More importantly, the authors choose TD3, an algorithm designed for continuous control problems (robotics, simulation), to tackle a discrete combinatorial assignment problem (matching drivers to orders). This is a mismatch: the natural RL tool for discrete action spaces is DQN or one of its variants. By shoehorning TD3 into this context, the authors create unnecessary complexity and introduce awkward mathematical approximations.
- The theoretical contribution of the paper is unclear and lack rigorous justification. The definition of z(·) is left as a black box, yet it is foundational to the method. Without clarity on this function, it’s impossible to know what policy the system is actually learning or whether its gradients are meaningful.
- The literature review is incomplete, missing key prior work. Several papers have addressed the similar problem, for example:
   - Qin, Z., Tang, X., Jiao, Y., Zhang, F., Xu, Z., Zhu, H., & Ye, J. (2020). Ride-hailing order dispatching at DiDi via reinforcement learning. INFORMS Journal on Applied Analytics, 50(5), 272-286.
   - Xu, Z., Li, Z., Guan, Q., Zhang, D., Li, Q., Nan, J., ... & Ye, J. (2018, July). Large-scale order dispatch in on-demand ride-hailing platforms: A learning and planning approach. In Proceedings of the 24th ACM SIGKDD international conference on knowledge discovery & data mining (pp. 905-913).
    The paper did not provide a comprehensive literature review to compare their paper with the existing ones. Nor did they compare their methods with these baselines.

---

> ### Author Rebuttal · Authors · 2025-07-30
>
> We are grateful for your thoughtful feedback. Your assessment is valuable for us to polish our paper. We are glad to address your questions.
>
> > Terminology Issue
> >
> > W1: The authors repeatedly use “delivery” terminology ...
> >
> > Q1: What is the research problem...
>
> **Re:** Thank you for pointing them out. We apologize for the misuse of terminology and typos in the paper. We will revise these according to your suggestions. Regarding the "decline order", you are correct that the subject should be customers, and we mistakenly referred to them as drivers. In the revised version, we will include definitions for terms such as "service rate" and "pickup time" in the experimental setup section of the appendix. However, we believe that these are mainly terminology issues, which will be corrected in the revised manuscript and will not fundamentally influence the scholarly merit of this paper. .
>
> > Deadheading Issue
> >
> > W2: Their model ignores the time and distance...
>
> **Re:** Thank you for your insightful comment. **Our method and simulation do take "pickup time" into account. (see Figure 3 and Figure 6f of experiment part)** To clarify this point, we will add a detailed description in the appendix under the experiment setup.
>
> > Customer Impatience Issue
> >
> > W3: There is no concrete modeling of...
>
> **Re:** Thank you for your insightful comment. In our experiment, we established a maximum waiting time of 5 minutes; after this period, the customer will decline the offer. We will include this description in the experimental setup section for clarity.
>
> > Algorithm Choice Issue
> >
> > W4: It is doubtful whether the authors are deeply versed...
> >
> > Q2: Please re-examine your algorithmic choices...
>
> **Re:** Thank you for your insightful comment. First, regarding the term SARL, we are not the first to use this terminology in the literature [1-2]. Our intention in adopting SARL is to differentiate our approach from existing RL-based ride-sharing methods, which predominantly utilize MARL. While we acknowledge that SARL is not a widely recognized term, we believe it helps clarify our focus in this paper, especially since both our SARL method and previous MARL methods fall under the umbrella of RL.
>
> Second, regarding the DQN-based ride sharing methods, they primarily support MARL settings, but often result in less efficient cooperation among drivers. In this paper, we aim to address this issue by developing a SARL-based method. The large action space (demonstrated in Appendix A) hinders the scalability and effectiveness of DQN-based methods. To tackle this, we propose a variant of TD3 with an action decomposition mechanism, which effectively realizes the SARL paradigm for ride-sharing. While the methodology may be complex, we believe it is necessary to manage the large action space. Additionally, we would like to draw the reviewer's attention to a series of works that utilize "continuous control" methods, including DDPG and TD3, to address dispatching and ride-hailing tasks [3-4].
>
> Additionally, we have already compared our methods with DQN-based approaches, such as DeepPool and BMG-Q in Section 4 and Appendix E, where the proposed method significantly outperfomed the DQN based approaches.
>
> > Theoretical Contribution Issue
> >
> > W5: The theoretical contribution of the paper...
>
> **Re:** Thank you for your insightful comment. In our method, the function 'z' maps the independent action probability of each driver to the joint action probability of the entire system (Eq. 5). We acknowledge that our initial illustration of 'z' was not clear enough, and we will provide a clear explanation in the revised version:
>
> For the centralized SARL method, we require the joint action probability of the entire driver group. However, the large action space makes it challenging to directly learn these probabilities for each joint action, complicating both action selection and sampling. To address this, we propose an action decomposition method that breaks down the joint action probability into the independent action probabilities of each driver. Actually, the independent action probability does not exist in a real sense. However, for any given definition of 'z', a corresponding independent action probability can be established. We then define our 'z' function as follows:
>
> **Eq. (I):**
> $$
> z(x) = ax^b
> $$
> where a,b,x>0. We believe this is a valid design since 'z' is an increasing function. Intuitively, if a driver-order pair has a higher probability of being chosen in the independent condition, it should have a higher utility and thus a higher probability of being selected in the joint action. This 'z' function possesses the advantageous property of maintaining a positive constant elasticity, allowing us to eliminate the elasticity term  \epsilon in Eq. 14 without affecting the gradient direction. As a result, when training our method using the gradient in Eq. 7, the definition of the 'z' function is implicit as given in Eq. (I). Given our method's SOTA performance, we consider this approach to be effective.
>
> > Baseline Issue
> >
> > W6: The literature review is incomplete...
> >
> > Q4: If you want to claim progress...
>
> **Re:** Thank you for your insightful suggestion. While we acknowledge that the two referenced works are indeed promising and pioneering in **ride-hailing**, we believe they do not fall within the scope of this **ride-sharing** paper. The two works utilize a similar paradigm that first estimates the V-value at each step and then calculates the advantage function as follows:
>
> **Eq. (II):**
> $$
> A(s,a) := \gamma^{\Delta t} V(s') + R(s,a) - V(s)
> $$
> They achieve dispatching through bipartite matching (ILP) to maximize the global advantage value. In their papers, the reward function for each action is predefined and equal to the order price. However, this reward design is unfeasible in ride-sharing contexts, as it does not account for the impact of detour time, leading to the delivery time of enroute customers infinite.
>
> Conversely, when we adapt our comprehensive reward to this papers, the decision-making time becomes excessively high. In our approach, the reward does not need to be computed during the decision phase. In contrast, the two referenced papers necessitate calculating the reward for each driver-order pair during decision-making. Since our reward function is tied to potential delivery time, a TSP must be solved to determine the reward for each pair. In our experiments, we found that for each simulation step (1 minute), this calculation time could reach 3 minutes, making this approach impractical. Given the real-time requirements, we conclude that these two ride-hailing works are not suitable for our ride-sharing task.
>
> Finally, regarding the operational methods mentioned by the reviewer, we believe the advantages of MARL have been well established by previous works. Most contemporary operational methods are based on the work of [5], which primarily focuses on optimizing the current step reward but struggles to capture long-term uncertainty (or requires extensive computation time, violating the real-time demands of ride-sharing dispatching). By experiment, we found RL-based methods do significantly outperform these operational methods, which is why we did not include them in the experimental section.
>
> In conclusion, we believe the current experiments sufficiently illustrate the superiority and efficiency of our methods, as we compare the latest SOTA approaches across different types of MARL, including CTCE, CTDE, and DTDE.
>
> > Q3: What is the definition of z(·)? ...
>
> **Re:** Thank you for your suggestion. Regarding the question about z(⋅), please refer to our response to the comment `W5`
>
> In the context of the ILP (bipartite matching) part, we utilize the Hungarian algorithm, which is included in the SciPy package in Python. This is a well-established and efficient method for finding the minimum weight matching in bipartite graphs. ILP methods have matured significantly and can solve the ILP problems associated with ride-sharing tasks very quickly (as evidenced by their application in the two works mentioned by the reviewer). In our experiments, we observed that decisions can be made within a second for each simulation step, achieving good real-time performance. We will add these details to the experimental setup section in the Appendix.
>
> For IDDQN, it stands for Independent Double DQN. We will include the full name in the revised paper.
>
> > Related Work and Title Issue
> >
> > Q5: Consider providing a dedicated related works...
> >
> > Q6: Consider changing the title of this paper...
>
> **Re:** Thank you for your insightful suggestion. In the revised version, we will include a dedicated section on related work in the appendix. Regarding the title, the term triple refers to both TD3 and the three BERTs. Since we cannot change the title at this stage, we intend to provide the answer directly within the paper: While the SARL approach can enhance cooperation and dispatching among drivers, we still require MARL for pre-training to mitigate data scarcity and provide a robust starting point.
>
> **References:**
>
> [1] Sarkar B, Talati A et al. Pantheonrl: A marl library for dynamic training interactions[C]//Proceedings of the AAAI Conference on Artificial Intelligence. 2022
>
> [2] Yu B, Lee T. Multi-agent reinforcement learning for the low-level control of a quadrotor UAV[C]//2024 American Control Conference (ACC). IEEE, 2024
>
> [3] Sivagnanam A, Pettet A et al. Multi-Agent Reinforcement Learning with Hierarchical Coordination for Emergency Responder Stationing[C]//International Conference on Machine Learning. PMLR, 2024
>
> [4] Wang Y, Wu J et al. Promoting collaborative dispatching in the ride-sourcing market with a third-party integrator[J]. IEEE Transactions on Intelligent Transportation Systems, 2024
>
> [5] Alonso-Mora J, Samaranayake S et al. On-demand high-capacity ride-sharing via dynamic trip-vehicle assignment[J]. Proceedings of the National Academy of Sciences, 2017

---

> > ### Comment · Reviewer_4BHb · 2025-08-05
> > **Additional minor comment**
> >
> > Thank you for the clarification. The paper would be clearer if the settings for pickup time and customer decline were introduced earlier. Additionally, could you elaborate on the differences between ride-hailing and ride-sharing? At present, it is challenging to assess the application-related contributions of this paper.
> >
> > Please update the version of the paper if possible.

---

> > > ### Author Response · Authors · 2025-08-05
> > >
> > > Thank you for your thoughtful feedback. In response to your suggestion, we will introduce the pickup time and customer decline settings earlier in the problem setting section to improve the clarity of our manuscript.
> > >
> > > Regarding the differences between ride-hailing and ride-sharing, ride-hailing generally refers to a service model in which a driver accepts only one passenger at a time, transporting them directly from their origin to their destination without sharing the vehicle with others whose journeys differ. There is no bundling of requests; each ride is point-to-point and exclusive to the booking party. In contrast, the ride-sharing scenario studied in our paper allows multiple passengers, each with distinct origins and destinations, to share the same vehicle simultaneously. This ride-sharing setting is significantly more complex than solo ride-hailing, as it requires the platform to make real-time decisions about which passengers to bundle together, how to dynamically route and reroute vehicles in response to newly arrived and en-route orders, and how to minimize detours and waiting times while considering the uncertainties of future requests. Our proposed framework directly addresses these challenges, providing a systematic solution for dynamic matching, dispatch, and routing that is necessary for practical ride-sharing operations.
> > >
> > > With regard to the application-related contributions of our work, we emphasize that efficient ride-sharing plays a crucial role in promoting convenient and sustainable urban transportation services. By enabling greater sharing among passengers, our method not only increases platform profitability and operational efficiency but also helps reduce total vehicle miles traveled and per-capita carbon emissions compared to solo rides. This, in turn, supports environmental sustainability goals. Moreover, our centralized reinforcement learning framework improves coordination among drivers, reduces delivery and detour times, and enables the platform to serve more orders within the same time frame. As a result, both platform income and customer satisfaction are enhanced, while also contributing to a greener and more efficient transportation system. We believe these contributions highlight the practical significance and societal value of our research.
> > >
> > > At this stage, the system does not permit us to upload a revised manuscript. However, we will promptly submit an updated version incorporating all the suggested improvements as soon as the system allows for resubmission.
> > >
> > > Thank you again for your constructive comments and for considering our work.

---

### Official Review · Reviewer_D4jT · 2025-07-10

**Clarity:** 3
**Significance:** 3
**Originality:** 3
**Rating:** 4
**Confidence:** 3

**Summary:**

On-demand ride-sharing platforms face real-time passenger-vehicle matching challenges amid uncertainties. The paper propose Triple-BERT, a centralized TD3-based method with action decomposition and a BERT network, to handle large-scale order dispatching. Validated on Manhattan data, it improves served orders by 4.26% and cuts pickup times by 22.25%.

**Questions:**

What is the basis for determining the order of location encoding for all orders and vehicles?

**Ethical Concerns:**

["NO or VERY MINOR ethics concerns only"]

**Quality:**

3

**Strengths And Weaknesses:**

Strengths
Using the BERT algorithm enables the paper to achieve remarkable conciseness.

---

> ### Author Rebuttal · Authors · 2025-07-30
>
> We are grateful for your thoughtful feedback. Your assessment is valuable for us to polish our paper. We are glad to address your question.
>
> > Q1: What is the basis for determining the order of location encoding for all orders and vehicles?
>
> **Re:**  Thank you for your insightful question. In our modified BERT model, we have removed the positional embedding, which allows all positions to be treated equally. Consequently, the model's output remains consistent, regardless of the order in which we encode the locations for all orders and vehicles.
>
> The rationale behind eliminating the positional embedding includes:
>
> **(1) Incorporated Position Information in State Space:** The coordinates of each vehicle and order are included as part of the state input, making additional positional embeddings unnecessary.
>
> **(2) Nature of Vehicle and Order Dispatching:** In ride-sharing tasks, the optimal assignment should be independent of the input sequence order. By removing the positional embedding, we ensure that all positions are homogeneous, thus realizing this property.
>
> **(3) Scalability Considerations:** Utilizing positional embeddings in BERT requires a predefined maximum input length before model training, which cannot be altered later. In scenarios of extreme concurrency, the number of orders may exceed this maximum length, potentially compromising model efficacy.
>
> We hope this explanation resolves your concerns.
>
> ------
>
> Finally, we express our heartfelt gratitude for your insightful questions and suggestions. They have been invaluable to our work and will be integral to our revised manuscript. **We hope our efforts will adequately address your concerns and that you will recognize our work.**

---

### Author Response · Authors · 2025-08-06
**Gentle Reminder: We Appreciate Your Further Comments**

Dear Reviewers, AC, SAC, and PC:

Thank you for your efforts and valuable feedback on our paper. As the discussion period will conclude in three days, we kindly request that any reviewers with additional comments or concerns share them with us. We would like the opportunity to address your questions before the deadline.

------

To facilitate your re-evaluation of our work, we have summarized the positive recognition of our contributions from all reviewers and how we have addressed the comments. We are encouraged by the following points:

1. **Meaningful Problem Background and Clear Motivation**: Our paper highlights the low efficiency and dimensional challenges faced by conventional MARL methods in ride-sharing platforms. (Reviewer 4Bhb, qVKq)

2. **Novel and Promising Framework**: We propose the first single-agent RL method for large-scale order dispatching in ride-sharing platforms, featuring a carefully designed neural network and training paradigm. (Reviewer D4jT, 4Bhb, qVKq)

3. **Extensive Large-Scale Experiments**: We compare our method with various MARL approaches, including DEDE, CTDE, and CTCE, using large-scale real-world datasets, demonstrating the SOTA performance of our method in all scenarios. (Reviewer D4jT, 9Qmh)

4. **Well-Organized Writing with Fully Open-source Code**: We provide a detailed methodology description and formula derivations in both the main paper and appendix, along with fully open-source code, processed data, and trained parameters, ensuring reproducibility. (Reviewer 4Bhb, qVKq)

We have addressed all reviewer comments and will incorporate them into the revised version, which includes the following updates:

1. **Expanded Ablation Study**: We have added experiments examining the influence of positional embeddings and attention-based encoders on our method. (Reviewer qVKq, 9Qmh)

2. **Expanded Generalization Evaluation**: We have included evaluation results across different days. (Reviewer 9Qmh)

3. **Expanded Scalability Evaluation**: We have added evaluation results for different methods under larger driver and order amounts, including running times.  (Reviewer qVKq)

These additional experiments further illustrate the superior performance of our method, particularly highlighting its high generalization and scalability, which suggest strong practical application potential. Additionally, we have followed the suggestions of Reviewers 4Bhb and 9Qmh to enhance the clarity of our writing.

------

Finally, if you are satisfied with our responses, we kindly request that you consider updating your scores to reflect the newly added results and discussions. We remain committed to addressing any remaining points you may have throughout the remainder of the discussion phase. We will be available to answer any questions about our paper until the end of the rebuttal phase, ensuring a comprehensive understanding. Thank you once again for your constructive comments and for considering our work.

 The Authors of Paper 10344

---

### Note · Authors · 2025-08-11

Dear Reviewers, AC, SAC, and PC:

Thank you for your valuable feedback on our paper, your considerable efforts during the review and discussion phases, and your dedication to the conference. We appreciate this opportunity to summarize our paper's contribution and the revisions made during the rebuttal period.

This paper presents **the first SARL method for ride-sharing**, departing from the conventional MARL paradigm. Our approach effectively addresses critical limitations of MARL-based solutions—such as the curse of dimensionality, poor coordination, and unstable training—while **achieving SOTA performance**. The method offers significant practical value: increased order fulfillment rates and reduced pickup/detour times not only enhance platform revenue and customer satisfaction but also decrease total vehicle miles traveled and per-capita carbon emissions.

------

During the rebuttal and discussion periods, we incorporated key reviewer suggestions to enhance both the experimental analysis and writing clarity:

1. **Expanded Ablation Study:** We added experiments analyzing the impact of positional embeddings and attention-based encoders. Combined with our original analysis of QK-attention, pre-training, and noise types, this provides a **comprehensive understanding of each module's contribution**.
2. **Large-Scale Scenario Validation:** We extended our evaluation to larger order volume and driver amount, demonstrating our method's **superior scalability and performance**. Crucially, it maintains **real-time decision speeds (within seconds)** suitable for minute-level operational cycles.
3. **Enhanced Generalization Analysis:** Building upon the original cross-period experiments, we conducted additional evaluations across diverse days. These results collectively confirm the method's **robust generalization capabilities**.
4. **Writing Revisions:** We have implemented all suggested improvements to the main text and appendix, enhancing clarity, terminology, and exposition. These revisions will be fully visible in the final version.

------

We think all reviewer concerns have been addressed, as we received no further queries following our latest responses. We hope this summary assists reviewers and ACs in their final deliberations. Once again, we sincerely thank you for your insightful comments on our work and your vital contributions to this conference.

Sincerely,

Authors of the paper 10344

---

### Decision · Program_Chairs · 2025-09-17

**Decision:**

Reject

**Comment:**

This paper proposes a centralized method designed specifically for large-scale order dispatching on ride-sharing platforms, and four reviewers have submitted their recommendations for the manuscript. This paper mitigates the limitation of conventional MARL-based methods, such as curse of dimensionality, poor coordination etc. However, main concerns for this work are the unrealistic experiment setup, unreliable independent assumptions etc. Reviewers have different opinions on this paper. During the discussion period, no one has argued for an acceptance.